# On the Origins of the Block Structure Phenomenon in Neural Network Representations

**Thao Nguyen**                                                       *thaottn@cs.washington.edu*
*Paul G. Allen Center for Computer Science & Engineering*
*University of Washington*[†]

**Maithra Raghu**                                                     *maithrar@gmail.com*
*Samaya AI*[†]

**Simon Kornblith**                                                   *skornblith@google.com*
*Google Research, Brain Team*

**Reviewed on OpenReview:** *https://openreview.net/forum?id=9tl6zjLYVS*

## Abstract

Recent work by Nguyen et al. (2021) has uncovered a striking phenomenon in large-capacity neural networks: they contain blocks of contiguous hidden layers with highly similar representations. This *block structure* has two seemingly contradictory properties: on the one hand, its constituent layers exhibit highly similar dominant first principal components (PCs), but on the other hand, their representations, and their common first PC, are highly dissimilar across different random seeds. Our work seeks to reconcile these discrepant properties by investigating the origin of the block structure in relation to the data and training methods. By analyzing properties of the dominant PCs, we find that the block structure arises from *dominant datapoints* — a small group of examples that share similar image statistics (e.g. background color). However, the set of dominant datapoints, and the precise shared image statistic, can vary across random seeds. Thus, the block structure reflects meaningful dataset statistics, but is simultaneously unique to each model. Through studying hidden layer activations and creating synthetic datapoints, we demonstrate that these simple image statistics dominate the representational geometry of the layers inside the block structure. We explore how the phenomenon evolves through training, finding that the block structure takes shape early in training, but the underlying representations and the corresponding dominant datapoints continue to change substantially. Finally, we study the interplay between the block structure and different training mechanisms, introducing a targeted intervention to eliminate the block structure, as well as examining the effects of pre-training and Shake-Shake regularization.

## 1 Introduction

Many modern successes of deep neural networks have adopted simple techniques to systematically increase model capacity, often through scaling architecture depth and width (Tan & Le, 2019). These large capacity models typically also maintain strong performance even in tasks with small amounts of training data. This has led to their widespread use across many different applications, including data-scarce, high-stakes settings such as medical imaging (Wang et al., 2016; Liu et al., 2017).

However, recent work has shown that representational structures of large capacity models trained on small datasets exhibit distinctive properties that are not present in shallower/narrower networks. Specifically,

---

[†]Work done primarily while at Google Research.

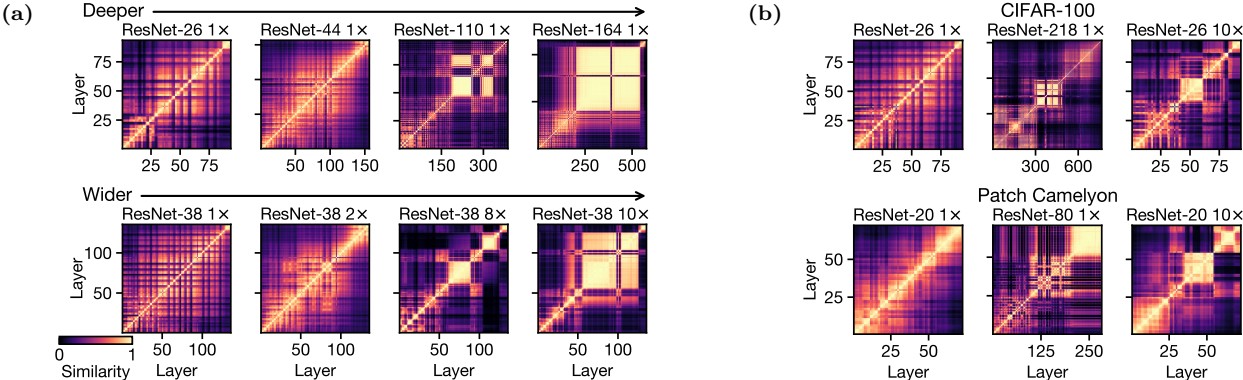

**Figure 1: (a): Block structure arises in wide and deep networks.** Heatmaps show similarity between layer representations, measured with linear CKA, for models of varying widths and depths trained on CIFAR-10. As models become wider or deeper, "blocks" of consecutive layers share similar representations. **(b): Block structure arises on many datasets.** Models trained on CIFAR-100 and the medical histopathology dataset Patch Camelyon show similar behavior to those in panel (a). Block structure also appears when networks trained on CIFAR-10 are evaluated on OOD images; see Appendix B.

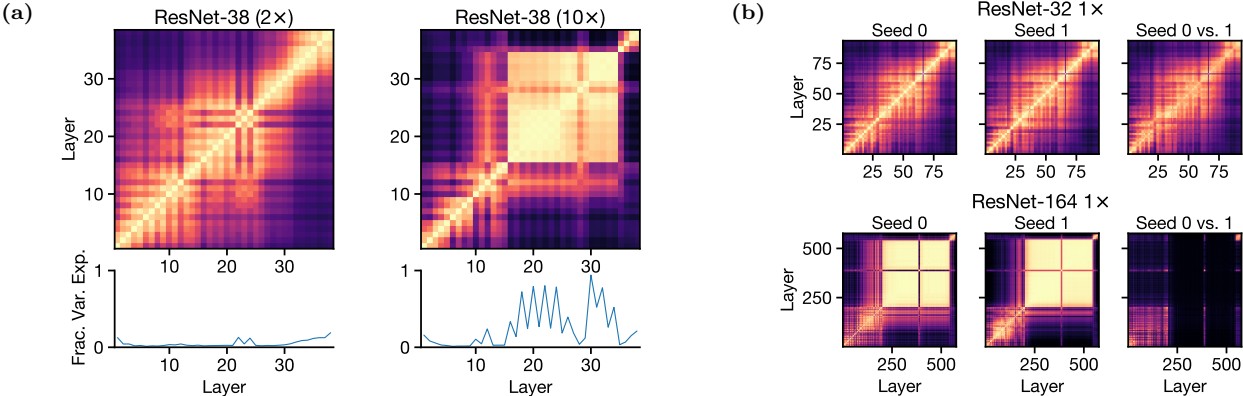

**Figure 2: (a): The first principal component of representations inside the block structure explains the majority of the variance in representations.** Top: Linear CKA similarity heatmaps for networks that do and do not exhibit block structure. Bottom: Fraction of variance in representations explained by first principal component. See Appendix C for similar findings with deep networks. **(b): Representations inside the block structure are highly unique to each seed.** 2 leftmost panels show similarity of representations *within* networks trained with 2 different random seeds. Rightmost panel shows similarity *between* the 2 networks.

when using linear centered kernel alignment (CKA) (Kornblith et al., 2019) to measure similarity between hidden representations of neural network layers, Nguyen et al. (2021) show that a large set of contiguous layers share highly similar representations. This is visible as a clear *block structure* in the heatmap of pairwise linear CKA similarities between layers (see Figure 1a). The block structure phenomenon is robust, appearing both in models trained on natural image datasets and those trained on medical imaging datasets (Figure 1b), as well as in a variety of CNN architectures (Nguyen et al., 2021; Cianfarani et al., 2022; Lu et al., 2021; Lange et al., 2022).

The emergence of the block structure with increasing model capacity is accompanied by significant changes in other properties of the individual layers. As shown in Figure 2a (left), in shallow/narrow networks, the first principal component of each layer exhibits relatively little variance. However, in higher-capacity networks (Figure 2a (right)), the layers that make up the block structure have dominant first principal components that explain most of the variance in their hidden representations. Naïvely, one might expect this collapse in rank to impair generalization (Daneshmand et al., 2020; Dong et al., 2021), but these networks nonetheless perform well on the distributions they are trained on. The representations of the layers that make up the

block structure are also dissimilar across seeds (Figure 2b), which is surprising in light of previous work suggesting that increased capacity and better generalization is typically associated with more consistent solutions across seeds (Morcos et al., 2018).

Thus, the block structure is arguably the single largest representational change associated with overparameterization, manifested through changes in similarity between layers as well as in the singular value spectra of individual layer representations. It consists of low-rank representations that are similar across layers, but differ across initializations. Having previously been associated with impaired generalization, these properties motivate a pressing question — is the block structure a sign of harmful overfitting to the data, or is it a harmless consequence of overparameterization? In this paper, we uncover answers to this question, exploring the origin of the block structure in relation to the data and training methods, and reconciling its contradictory behaviors. Specifically, our results are as follows:

- The dominant first principal components of the layers that make up the block structure arise from a small number of *dominant datapoints* that share similar characteristics (e.g. background color). Like the block structure phenomenon, dominant datapoints are found in large-capacity neural networks.

- The set of dominant datapoints (and their shared characteristics) can vary across training runs, which explains the observed block structure dissimilarities across seeds.

- The block structure is caused by these dominant datapoints. When dominant datapoints are excluded from the dataset used for representational analysis, the block structure disappears.

- Dominant datapoints produce high activation norms inside the hidden layers corresponding to the block structure. By constructing synthetic examples based off of the shared image characteristics of the dominant datapoints, we show that these image characteristics are indeed responsible for the high activation norms.

- The block structure emerges early in training, but early block structures have different representations and yield different dominant datapoints than the final block structure found at the end of training.

- We show that it is possible to eliminate the block structure from the internal representations using a novel principal component regularization method. However, this intervention provides only minor accuracy improvements, suggesting that the block structure has, at most, a minor impact on networks' generalization abilities.

## 2 Related Work

Previous work has studied certain propensities of deep neural networks in the standard training setting, such as their simplicity bias (Huh et al., 2021; Valle-Perez et al., 2018; Nakkiran et al., 2019), texture bias (Baker et al., 2018; Geirhos et al., 2018; Hermann et al., 2019) and reliance on spurious correlations (McCoy et al., 2019; Geirhos et al., 2020; Ribeiro et al., 2016; Jo & Bengio, 2017; Hosseini & Poovendran, 2018). Inspired by the findings of Nguyen et al. (2021) that deep and wide networks learn many layers with similar representations, we seek to characterize what signals these layers convey and may be overfitting to. To do so, we analyze the behavior of the *internal representations* of models of varied depths and widths, using methods for measuring similarity of neural network hidden representations (Kornblith et al., 2019; Raghu et al., 2017; Morcos et al., 2018), as well as standard tools of linear algebra. Representational similarity techniques have previously shed light on model training procedures (Gotmare et al., 2018; Neyshabur et al., 2020), features (Resnick et al., 2019; Thompson et al., 2019; Hermann & Lampinen, 2020), and dynamics (Maheswaranathan et al., 2019), and has also furthered understanding of network internals in applications of deep learning, such as medicine (Raghu et al., 2019) and machine translation (Bau et al., 2019).

Our work also relates to previous attempts to understand the properties of overparameterized models (Zhang et al., 2016). Theoretical work in this area has focused on linear models, models with random features, or kernel settings (Belkin et al., 2018; Hastie et al., 2019; Liang et al., 2020; Bartlett et al., 2020), all of which lack intermediate features, or involves linear networks (Advani et al., 2020). Our results suggest that the behavior of intermediate features of practical neural networks changes dramatically with increasing overparameterization, in ways that are not obvious from previous analysis, and cannot be easily characterized by properties at initialization.

## 3 Experimental Setup and Background

**Measuring Representation Similarity with CKA:** Centered kernel alignment (CKA) (Kornblith et al., 2019; Cortes et al., 2012) addresses several challenges in measuring similarity between neural network hidden representations including (i) their large size; (ii) neuronal alignment between different layers; and (iii) features being distributed across multiple neurons in a layer. Like Nguyen et al. (2021), we use the minibatch implementation of linear CKA with a batch size of 256 sampled without replacement from the test dataset, and accumulate statistics over 10 epochs to allow the minibatch estimator to converge.

More concretely, given $k$ minibatches of $n$ examples, and two layers having $p_1$ neurons and $p_2$ neurons each, minibatch CKA takes as inputs $k$ pairs of centered activation matrices $(\boldsymbol{X}_1, \boldsymbol{Y}_1), \ldots, (\boldsymbol{X}_k, \boldsymbol{Y}_k)$ where $\boldsymbol{X}_i \in \mathbb{R}^{n \times p_1}$ and $\boldsymbol{Y}_i \in \mathbb{R}^{n \times p_2}$ reflect the activations of these layers to the same minibatches. It produces a scalar similarity score between 0 and 1 by averaging the scores of Hilbert-Schmidt independence criterion (HSIC), computed with a linear kernel, over the minibatches:

$$\text{LCKA}_{\text{minibatch}} = \frac{\frac{1}{k} \sum_{i=1}^{k} \text{HSIC}_1(\boldsymbol{X}_i \boldsymbol{X}_i^\mathsf{T}, \boldsymbol{Y}_i \boldsymbol{Y}_i^\mathsf{T})}{\sqrt{\frac{1}{k} \sum_{i=1}^{k} \text{HSIC}_1(\boldsymbol{X}_i \boldsymbol{X}_i^\mathsf{T}, \boldsymbol{X}_i \boldsymbol{X}_i^\mathsf{T})} \sqrt{\frac{1}{k} \sum_{i=1}^{k} \text{HSIC}_1(\boldsymbol{Y}_i \boldsymbol{Y}_i^\mathsf{T}, \boldsymbol{Y}_i \boldsymbol{Y}_i^\mathsf{T})}}, \tag{1}$$

where $\text{HSIC}_1$ is the unbiased estimator of HSIC from Song et al. (2012). This estimator of linear CKA converges to the same value regardless of the batch size.

We compute CKA between *all* layer representations, including before and after batch normalization, activations, and residual connections. In experiments that involve tracking how a model's internal properties (principal components of activations, representation similarity, etc.) change across epochs, we set batch normalization layers to be in training mode, to reduce the difficulty of adapting to batch statistics of the test set when the model has not converged.

**The Block Structure Phenomenon:** Nguyen et al. (2021) use linear CKA to compute the representation similarity for all pairs of layers within the same model and visualizes the result as a heatmap (with x and y axes indexing the layers from input to output). They find a contiguous range of hidden layers with very high representation similarity (yellow squares on heatmaps in Figure 1a) in very deep or wide models, and call this phenomenon the block structure. The block structure arises in networks that are large *relative* to the size of the training set — while small networks may not exhibit block structure when trained on CIFAR-10, they do exhibit block structure on smaller datasets.

Representations of the layers making up the block structure exhibit different representational geometry than the rest of the layers. For layers inside the block structure, the first principal component explains a large fraction of the variance in representations; this is not the case for the other layers or for networks without the block structure (Nguyen et al., 2021). We replicate this observation in Figure 2a. The similarity between layers inside the block structure reflects the alignment of their first principal components, as can be seen from the following decomposition of linear CKA for centered activation matrices $\boldsymbol{X} \in \mathbb{R}^{n \times p_1}$, $\boldsymbol{Y} \in \mathbb{R}^{n \times p_2}$:

$$\text{LCKA}(\boldsymbol{X}, \boldsymbol{Y}) = \frac{\sum_{i=1}^{p_1} \sum_{j=1}^{p_2} \lambda_X^i \lambda_Y^j \langle \boldsymbol{u}_X^i, \boldsymbol{u}_Y^j \rangle^2}{\sqrt{\sum_{i=1}^{p_1} (\lambda_X^i)^2} \sqrt{\sum_{j=1}^{p_2} (\lambda_Y^j)^2}} \tag{2}$$

where $\boldsymbol{u}_X^i \in \mathbb{R}^n$ and $\boldsymbol{u}_Y^i \in \mathbb{R}^n$ are the $i^{\text{th}}$ normalized principal components of $\boldsymbol{X}$ and $\boldsymbol{Y}$, and $\lambda_X^i$ and $\lambda_Y^i$ are the amounts of variance that these principal components explain (Kornblith et al., 2019). As the fraction of variance explained by the first principal component of each representation approaches 1, CKA becomes a measure of the squared cosine similarity between first principal components $\langle \boldsymbol{u}_X^1, \boldsymbol{u}_Y^1 \rangle^2$. Nguyen et al. (2021) conclude that the block structure preserves and propagates a dominant first principal component across many hidden layers.

**Datasets & Models:** Our setup closely follows that of Nguyen et al. (2021) and analyzes ResNets of varying depths and widths, trained on common image classification datasets CIFAR-10 and CIFAR-100 (Krizhevsky et al., 2009), as well as the medical imaging dataset Patch Camelyon (Veeling et al., 2018). These datasets are chosen to reflect the image statistics found in different domains, and all easily induce a block structure in reasonably sized ResNets.

The ResNet architecture design follows Zagoruyko & Komodakis (2016), with the layers distributed evenly between three stages — each marked by a different feature map size — and the number of channels is doubled after each stage. To scale the model depth and width, we increase the number of layers and channels respectively. In experiments involving Shake-Shake regularization (Gastaldi, 2017), the network is modified to have 3 branches that are combined in a stochastic fashion. More information on hyperparameters can be found in Appendix A.

## 4 Dominant Datapoints and How They Shape the Block Structure

### 4.1 Datapoints that Activate the First PC

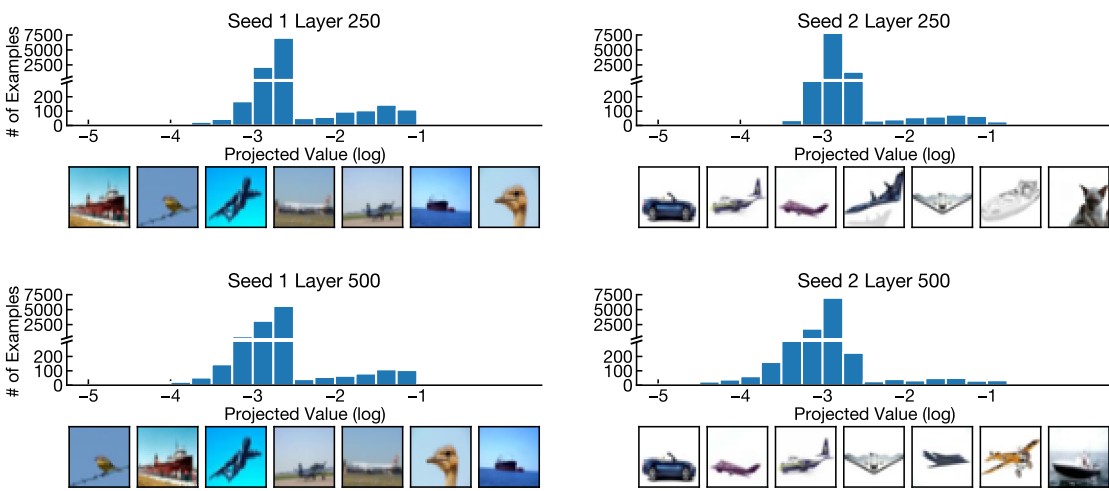

**Figure 3: Visualization of the distribution of projected values onto the first principal component by CIFAR-10 test inputs**. There exist a small number of datapoints that yield significantly larger projected values than the rest and dominate the first principal component of the network. Here we show examples of those dominant images for two different seeds of ResNet-164 (1×) (columns). Each seed's dominant images share similar background colors, but these background colors differ between seeds. Further visualization of dominant datapoints across different layers — for instance, layers 250 and 500 in this case — show that they are consistent across layers making up the block structure. See Appendix D for analysis of other models and tasks.

Motivated by previous evidence that the block structure propagates a dominant principal component across its constituent layers, we examine the distribution of the projections of each example's activations onto the first principal component. We find that the distribution is bimodal. Most examples have small projections, but some are orders of magnitude larger than the median (Figure 3).

We call these datapoints with large projections *dominant datapoints*, and find that they are consistent across the range of layers making up the block structure, as seen from each column of images in Figure 3 showing the dominant datapoints for two different layers in the same block structure. This explains why the first principal components of different layer activations inside the block structure are highly similar (Figure 2a), an observation made earlier in (Nguyen et al., 2021). Moreover, this dominant datapoint phenomenon is present only in networks that also exhibit a block structure. As shown in Appendix Figure 15, in networks without block structure, projections on the first principal component are unimodally distributed and the corresponding datapoints differ between layers.

Dominant datapoints are visually similar. In the left column of Figure 3, we observe that all datapoints have a blue background, although the precise shade of blue varies. However, the visual signals that the first principal component picks up on depend on the random seed used to train the model. The right column of Figure 3 shows the corresponding properties of an architecturally identical model trained from a different seed, where the dominant datapoints share white backgrounds instead. We observe that dominant datapoints also arise

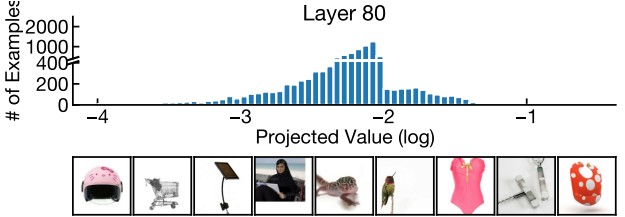

**Figure 4: Visualization of the distribution of projected values onto the first principal component by test inputs, for ResNet-50 (8×) trained on 1/16 of ImageNet.** Top row shows histograms of the projected values on the first PC. Bottom rows show images with the largest projections on the first PC.

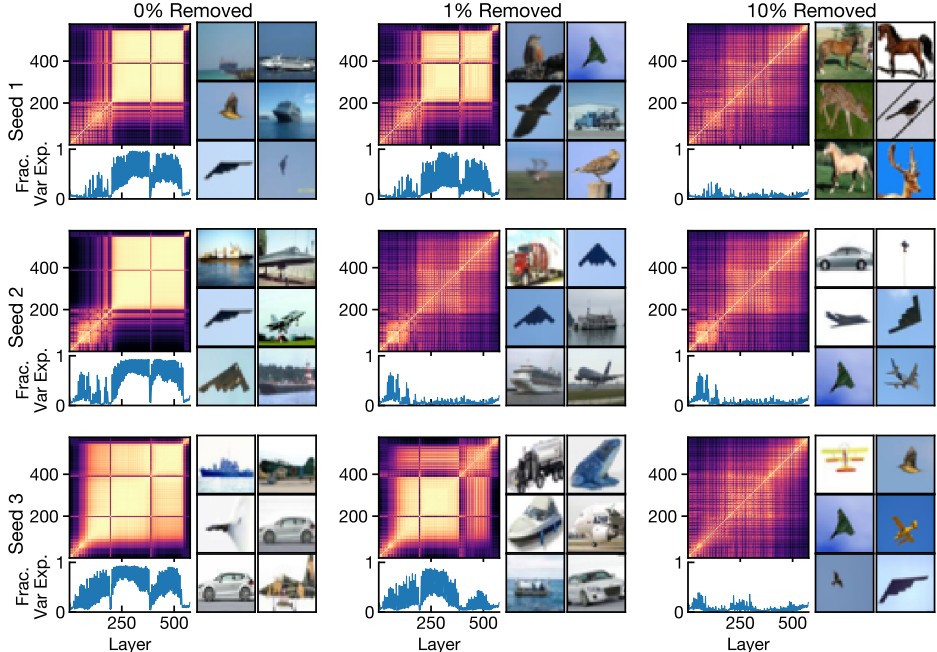

**Figure 5: Removing a small number of dominant datapoints eliminates the block structure.** Plots show the effect of removing examples with the largest projections on the first PC of layer 300 of ResNet-164 (1 ×) models trained on CIFAR-10. Columns reflect different numbers of examples removed; rows reflect models trained from different seeds. Within each group, the top left plot shows linear CKA heatmaps, the bottom left shows the fraction of variance explained by the first PC, and the images reflect the new examples with the largest projection on the first PC after data removal.

in ImageNet-trained ResNets (Figure 4). Refer to Appendix D for visualizations of dominant images found in other tasks (CIFAR-100, Patch Camelyon) and model architectures (wide ResNet). Besides background color, the dominant datapoints can also reflect other simple image patterns that are prevalent in the dataset, such as the appearance of large dark spots in histopathologic scans (Appendix Figure 17).

Finally, the block structure observed in linear CKA heatmaps arises solely from dominant datapoints. As seen in Figure 5, when the 10% most dominant datapoints are excluded from evaluation, the block structure is completely eliminated in all 3 training runs of ResNet-164 (1×) that we examined, and the fraction of variance explained by the first PCs is substantially reduced. In fact, for one training run, removing only the 1% most dominant datapoints is sufficient to achieve this effect. Thus, the block structure is completely determined by the dominant images, and is sensitive to the frequency of the dataset statistics that it captures.

## 4.2 Dominant Datapoints Have Large Activation Norms

We investigate what happens to the activations of a dominant example as it propagates through the network, and observe that it strongly activates the parts of the network with a visible block structure. Figure 6 shows two set of dominant datapoints, for a ResNet trained on CIFAR-10 that has a preference for white background images (top row), and for another ResNet trained on Patch Camelyon that responds to clear pink backgrounds

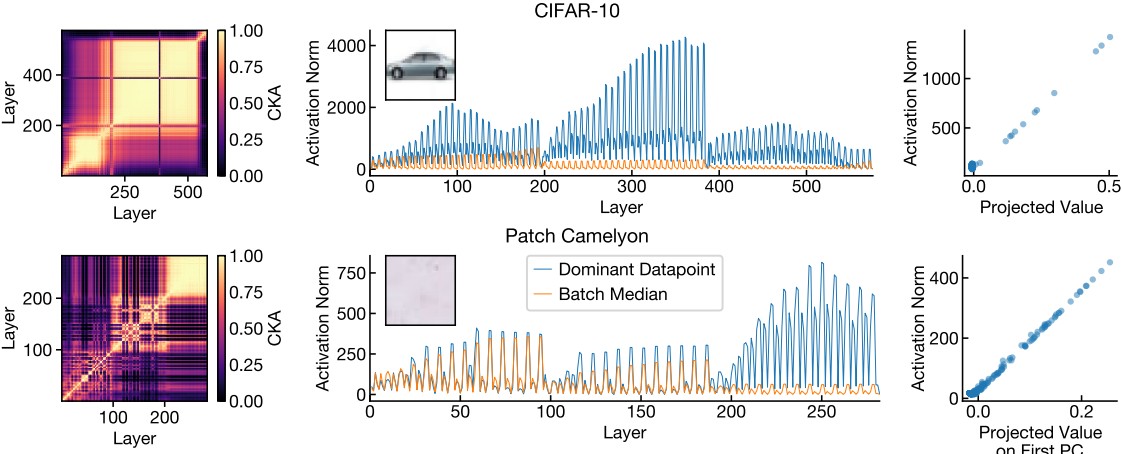

**Figure 6: Datapoints that dominate the first principal components of the block structure also strongly activate the corresponding layers**. We explore the relationship between dominant datapoints and activation norms for ResNet-164 (1×) trained on CIFAR-10 (top row) and ResNet-80 (1×) trained on Patch Camelyon data (bottom row). For layers inside the block structure (left column), dominant datapoints (inset) produce much larger activation norms than the median of a randomly selected minibatch (middle column). Moreover, within these layers, the norms of the activations of different datapoints are highly correlated with the magnitudes of their projections on the first principal component (right column).

(bottom row). Both models contain block structure in their internal representations, and we find that in the corresponding layers, the activations of the dominant datapoints are substantially larger in norm than the median activations of the minibatches they are a part of. Moreover, the magnitude of the projection on the first PC is correlated with the activation norm. We conclude that dominant datapoints evoke activations with large norms, and activations of different dominant datapoints point in similar directions.

In Appendix J, we include additional results measuring representational similarity in networks with block structure using CKA with different kernels. All kernels we have tested are sensitive to differences between dominant and non-dominant datapoints, but the linear kernel produces particularly strong "blocks" compared to RBF or cosine kernels. Across all kernels, the removal of dominant datapoints consistently eliminates blocks in representational similarity heatmaps.

### 4.3 Image Backgrounds As a Dominant Property

In this section, through data and training manipulations, we further characterize how dominant datapoints arise from specific colors in images.

First, we examine the connection between the background colors of dominant datapoints, which vary across random seeds, and layer activation norms. We begin with dominant examples and repeat only the top left pixel in each image across the entire dimensions of the image, obtaining solid color images. These synthetic images indeed yield even larger activations compared to the dominant examples they are taken from, and different initializations of architecturally identical networks respond to different synthetic images. For instance, given the ResNet-164 (1×) model that has dominant datapoints containing a blue background (Figure 3), its hidden layers are further activated when all image pixels are replaced with the same shade of blue, but a solid white image produces considerably smaller activations (Figure 7). In the same figure, we observe the opposite trend for another ResNet-164 (1×) seed, which has been shown to pick up on white backgrounds (see Figure 3). Refer to Appendix E for similar analysis on Patch Camelyon.

**Intervention: Color Augmentation:** Given earlier insights, a natural intervention to prevent the network from potentially picking up background color signal is adding color augmentation. This includes randomly dropping color channels and jittering brightness, contrast, saturation, and hue of training images (Howard,

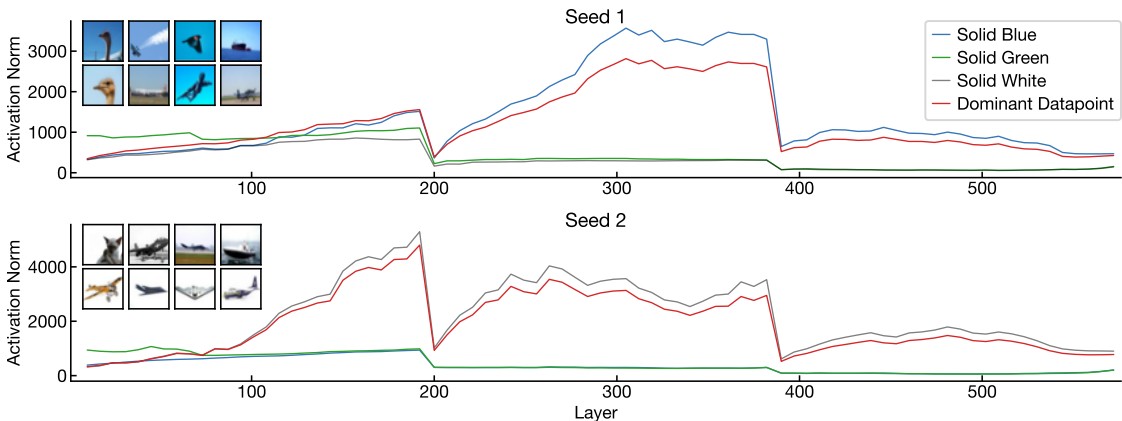

**Figure 7: Solid color images strongly activate intermediate layers**. Rows show ResNet-164 (1×) models that are trained from different random initializations. The top model's dominant datapoints consist of images with blue backgrounds (see inset images), whereas the bottom model prefers white background images. Layers of the top model are strongly activated by solid blue images, but not solid white images, whereas the bottom model shows the opposite pattern. Layers of both models are strongly activated by their respective dominant datapoints (red lines), but other solid colors (e.g. green) do not yield strong activations in either. To improve readability of the plot, we plot only the representations at the end of each ResNet block. See Appendix E for similar findings on Patch Camelyon models.

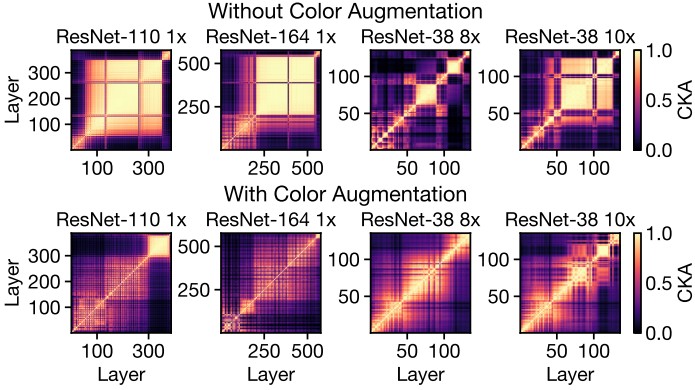

**Figure 8: Simply adding color augmentation helps reduce the block structure effect**. Having established that the activations and representational components of some large-capacity models pick up on common background colors from the inputs, we experiment with color dropping and color jittering during training to counter this effect. Indeed, color augmentation minimizes the block structure appearance in the internal representations.

2013; Szegedy et al., 2015). As shown in Figure 8, training with this data augmentation reduces the block structure in large capacity models.

## 5 Evolution of Block Structure during Training

In the previous section, we characterize the signals the block structure propagates across its layers, and explore their implications on other aspects of the network internals. Informed by these findings, we next explore what happens to the block structure and the dominant images over time, from initialization until the model converges, and how this process varies across different training runs.

Figure 9 shows the evolution of the internal representations of a ResNet-110 (1×) model as it is trained for 300 epochs on CIFAR-10, and tracks how similar each checkpoint is to the final model. We observe that some structure in the CKA heatmap is already present by the first epoch, and the heatmap undergoes little qualitative change past epoch 20 (top set of plots). However, when we inspect the corresponding dominant datapoints and compare the hidden representations between intermediate checkpoints and the final model (bottom rows of plots), we find that the block structure does not always carry the same information. Instead, the representations, and corresponding groups of dominant datapoints, only stabilize much later in training.

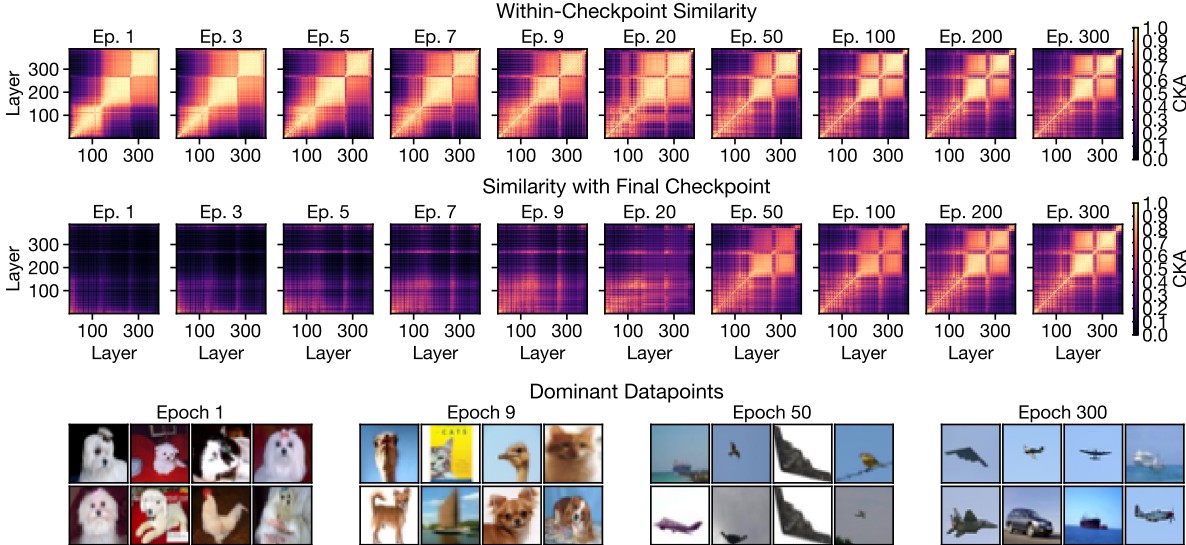

**Figure 9: Block structure phenomenon arises early during training, but the corresponding dominant datapoints continue to change substantially**. We compute the CKA between all pairs of layers within a ResNet-110 (1×) model at different stages of training, and find that the shape of the block structure is defined early in training (top row). However, comparing these different model checkpoints to the fully-trained model reveals that the block structure representations at different epochs are considerably dissimilar to the final representations, especially during the first half of the training process (middle row). The corresponding dominant datapoints also vary significantly over the course of training, even after the block structure is clearly visible in the heatmaps (bottom row). See Appendix F for similar plots with greater granularity, different seeds and architectures.

We observe similar behavior for other models in Appendix F, and note that the differences in block structure representations across random seeds already take shape near the start of training as well. Overall these findings suggest that the uniqueness of the block structure representations in large-capacity models can be attributed to both initialization parameters and the image minibatches the models receive throughout training.

To further explore the link between dominant datapoints and fluctuations in network representations, in Figure 10, we track the magnitude of the projected value on the first PC of a single dominant datapoint found at the *end* of training, and find that the value plummets at epochs when the internal representation structure diverges from that of the fully trained model. At these epochs, the dominant datapoint does not produce large activations either (bottom set of plots). This illustrates that the precise set of dominant datapoints can vary significantly over the course of training for large-capacity models.

## 6  Block Structure and Training Mechanisms

Having observed how the internal representation structures — specifically, the dominant PCs of layer representations — could vary significantly during training, we turn to examining the interplay between the block structure and the training mechanism. Although the block structure arises naturally with standard training, previous work has suggested that the block structure may be an indication of redundant modules in the corresponding networks (Nguyen et al., 2021). Thus, it is natural to ask whether it is possible to train large-capacity models without a block structure, and how such models perform compared to those with block structures.

Since the block structure reflects the similarity of a dominant first principal component, propagated across a wide range of hidden layers (see Section 3), we study whether regularizing the first principal components of layer activations would eliminate the block structure. More specifically, we estimate the fraction of variance explained by the first principal component of each layer using power iteration and penalize it in

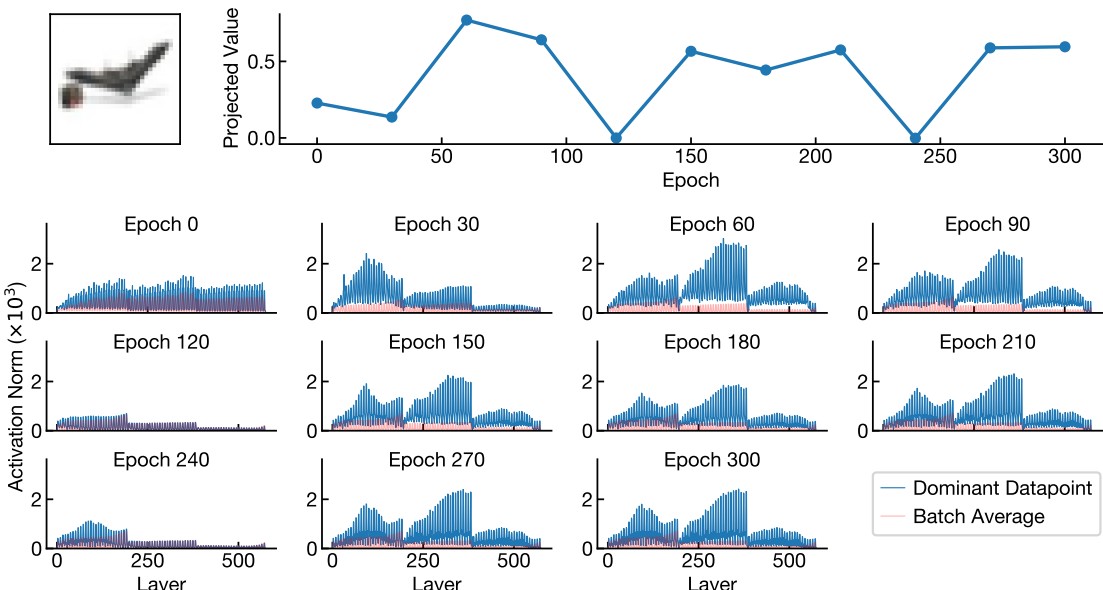

**Figure 10: Example of how the magnitudes of layer activations and the projected values onto the first principal component for a dominant image vary across different epochs, for a ResNet-164** (1×). Given a dominant datapoint for a ResNet-164 (1×) model, we track the magnitude of its projected value onto the first principal component of the block structure representations (top left), as well as its activation norms at each layer in the network (bottom set of plots), over time. Notice the correspondence between these 2 metrics, especially when their values drop at epochs 0 (initialization), 120 and 240. This is also aligned with the measurement of CKAs across different epochs (see Figure 22, where we find that the model checkpoints at these 3 epochs are highly dissimilar from the final model in terms of the hidden representations). See Appendix Figure 24 for the corresponding visualization of a dominant example of a wide ResNet (ResNet-38 (10×)).

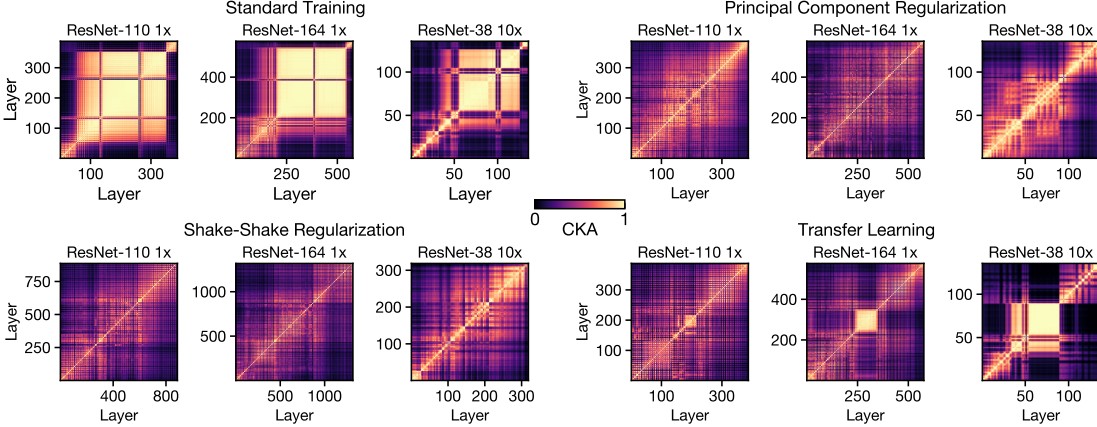

**Figure 11: Training with principal component regularization, transfer learning and Shake-Shake regularization helps eliminate the block structure**. We directly regularize the first PC of each layer activations given that this component explains a large fraction of variance in block structure representations, and find that this eliminates the block structure. Full algorithm details can be found in Appendix G. Shake-Shake regularization (Gastaldi, 2017) has a similar effect. We also find that transfer learning reduces the appearance of the block structure, although it is still present in the largest network. These results demonstrate that the block structure phenomenon is dependent on the training mechanism. See Appendix I for implications of these training methods on representations across random seeds.

the training objective when it exceeds 20%. We provide full implementation details in Appendix G. The resulting heatmap, in Figure 11 top right, shows that not only does this eliminate the block structure from the internal representations, but there is also no detrimental effect on performance (Table 1). We even observe

| Depth | Width | Accuracy (%) (standard training) | Accuracy (%) (PC regularization) |
|---|---|---|---|
| *CIFAR-10 subsampled (6% of the full dataset):* | | | |
| 56 | 1 | 77.8 ± 0.429 | **79.2 ± 0.188** |
| 26 | 8 | 80.1 ± 0.354 | **81.1 ± 0.185** |
| 26 | 10 | 80.3 ± 0.306 | **81.2 ± 0.194** |
| 38 | 8 | 80.2 ± 0.362 | **80.9 ± 0.264** |
| 38 | 10 | 80.3 ± 0.412 | **81.4 ± 0.350** |
| *CIFAR-10:* | | | |
| 110 | 1 | 94.3 ± 0.078 | **94.4 ± 0.063** |
| 164 | 1 | 94.4 ± 0.075 | **94.5 ± 0.063** |
| 26 | 10 | 95.8 ± 0.087 | **96.0 ± 0.051** |
| 38 | 8 | 95.7 ± 0.091 | **95.8 ± 0.080** |
| 38 | 10 | 95.7 ± 0.157 | **95.9 ± 0.067** |
| *CIFAR-100:* | | | |
| 218 | 1 | 74.1 ± 0.310 | **75.1 ± 0.132** |
| 224 | 1 | 74.0 ± 0.350 | **75.2 ± 0.131** |
| 38 | 8 | 79.8 ± 0.149 | **80.6 ± 0.306** |
| 38 | 10 | 80.5 ± 0.174 | **81.1 ± 0.241** |

**Table 1: Comparison of performance of large-capacity models on CIFAR-10 and CIFAR-100, with and without principal component regularization.** We observe that our proposed principal component regularizer consistently yields accuracy improvements for large capacity models that contain the block structure. The performance gains are particularly significant in the case of CIFAR-100 and subsampled CIFAR-10 datasets. Numbers are calculated based on 10 training runs.

| Depth | Width | Accuracy (%) (standard training) | Accuracy (%) (Shake-shake) | Accuracy (%) (transfer learning) |
|---|---|---|---|---|
| 110 | 1 | 94.3 ± 0.078 | **94.5 ± 0.051** | 94.4 ± 0.080 |
| 164 | 1 | 94.4 ± 0.075 | **94.6 ± 0.066** | 94.5 ± 0.102 |
| 26 | 10 | 95.8 ± 0.087 | **95.9 ± 0.024** | 95.9 ± 0.099 |
| 38 | 8 | 95.7 ± 0.091 | **95.9 ± 0.033** | 95.8 ± 0.024 |
| 38 | 10 | 95.7 ± 0.157 | **96.0 ± 0.106** | 95.8 ± 0.033 |

**Table 2: Comparison of performance of large-capacity models on CIFAR-10, between standard training and training with Shake-shake regularization and transfer learning.** We observe that the latter training techniques (last 2 columns) yield higher accuracies compared to standard training, especially in the case of Shake-shake regularization. Numbers are calculated based on 10 training runs.

small accuracy improvements on CIFAR-100 and in the low-data regime, as shown in Table 1. Although we designed this regularizer explicitly to eliminate the block structure, it is possible that it has other effects on the training dynamics that lead to this small accuracy improvements. Overall these experiments demonstrate that the detrimental impact of the block structure on generalization is at most minor.

Other standard training practices that are commonly used to boost performance (Table 2) are also effective at reducing or eliminating the block structure effect. Shake-Shake regularization (Gastaldi, 2017) eliminates the block structure for all of the network sizes that we examine (Figure 11, bottom left), whereas transfer learning (Figure 11, bottom right) and training with smaller batch sizes (Appendix H) generally reduce the appearance of the block structure, although blocks are still discernible in the largest models that we trained. In addition to regularizing the block structure, these training methods also produce more similar representations across different training runs of the same architecture configuration (Figure 12, Appendix Figure 27).

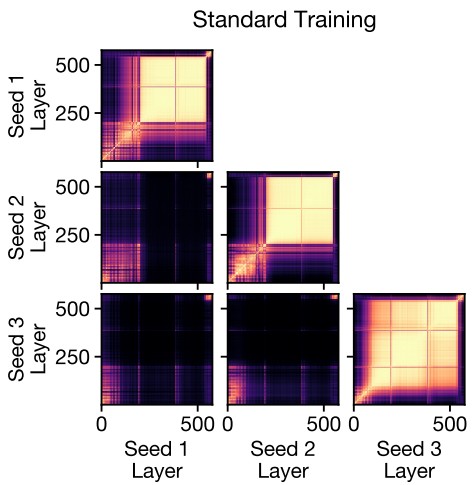
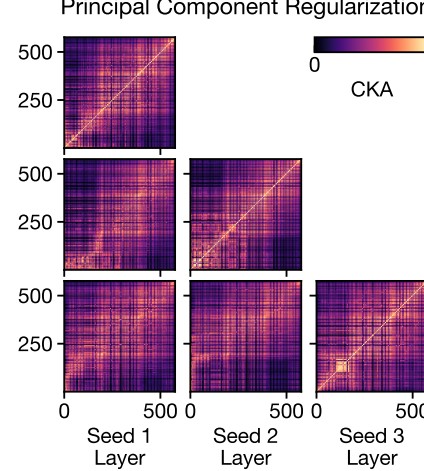

**Figure 12: Training with principal component regularization yields models that are more similar across different training runs**. Each group of plots shows CKA between layers of models with the same architecture but different initializations (off the diagonal) or within a single model (on the diagonal). Similar to the observation made in Figure 27, while representations across models are highly dissimilar in the standard training case, models trained with principal component regularization from different random initializations show more representational similarity in corresponding layers.

Overall, our findings suggest that it is possible to obtain good generalization accuracy in networks with and without block structure. We observe that learning processes that reduce the dominance of the first PC of the representations provide slightly higher accuracies than standard training. However, some caution is warranted in interpreting these performance benefits: it may be difficult to causally determine the connection between the block structure and performance, as any training intervention targeting the block structure may have other distinct ramifications that also affect performance.

# 7 Discussion

**Scope and Limitations:** Our work primarily focuses on the behavior of large-capacity networks trained on relatively small datasets. This is motivated by domains such as medical imaging where data is expensive relative to the cost of training a large model, and the high-stakes nature makes it important to understand the model's behavior. In general, understanding how the representational properties change with model capacity (relative to dataset size) is of both scientific and practical interest, as model size continues to grow over the years but there are many domains beyond vision (e.g. see Kaggle) where dataset size does not. Additional exploration is needed to study state-of-the-art settings in e.g. NLP, which use much bigger and heterogeneous datasets.

**Conclusion:** The block structure phenomenon uncovered in previous work (Nguyen et al., 2021) reveals significant differences in the representational structures of overparameterized neural networks and shallower/narrower ones. However, it also exhibits some contradicting behaviors — being unique to each network while propagating a dominant PC across a wide range of layers — that respectively suggest the underlying representations could either overfit to noise artifacts or capture relevant signals in the data. Our work seeks to provide an explanation for this discrepancy. We find that despite the inconsistency of the block structure across different training runs, it arises not from noise, but real and simple dataset statistics such as background color. We further discover a small set of dominant datapoints (with large activation norms) that are responsible for the block structure. These datapoints emerge early in training and vary across epochs, as well as across random seeds. We show how different training procedures, including color augmentation, transfer learning, Shake-Shake regularization, and a novel principal component regularizer, can reduce the influence of these dominant datapoints, eliminating the block structure and leading to more consistent representations across training runs.

Since the block structure phenomenon has been shown to robustly arise in large-capacity networks under various settings (Nguyen et al., 2021), rigorously characterizing its cause and effects is of great importance to understanding the nuances in the way these networks learn, despite their similarly good performances. We believe that insights into the representational properties of overparameterized models can benefit techniques that make direct use of the internal representations, such as transfer learning and interpretability methods. This work also motivates interesting open questions including exploring how dominant datapoints are manifested in other domains and applications of deep learning, as well as applying principal component regularization to distribution shift and self-supervision problems.

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

# Appendix

## A  Training Details

For wide ResNets, we look at models with depths of 14, 20, 26, and 38, and width multipliers of 1, 2, 4, 8 and 10. For deep ResNets, we experiment with depths 32, 44, 56, 110 and 164. In CIFAR-100 experiments, the block structure only appears at a greater depth so we also include depths 218 and 224 in our investigation. For Patch Camelyon datasets, we find that depth 80 is enough to induce a block structure in the internal representations. All ResNets follow the architecture design in (He et al., 2016; Zagoruyko & Komodakis, 2016).

Unless otherwise specified, we train all the models using SGD with momentum of 0.9 for 300 epochs, together with a cosine decay learning rate schedule and batch size of 128. Learning rate is tuned with values [0.005, 0.01, 0.001] and $L_2$ regularization strength with values [0.001, 0.005]. For CIFAR-10 and CIFAR-100 experiments, we apply standard CIFAR-10 data augmentation consisting of random flips and translations of up to 4 pixels. With Patch Camelyon, we use random crops of size 32x32, together with random flips, to obtain the training data. At test time, the networks are evaluated on central crops of the original images. For CKA analysis, each architecture is trained with 10 different seeds and evaluated on the full test set of the corresponding domain.

## B  Block Structure on Out-of-Distribution Data

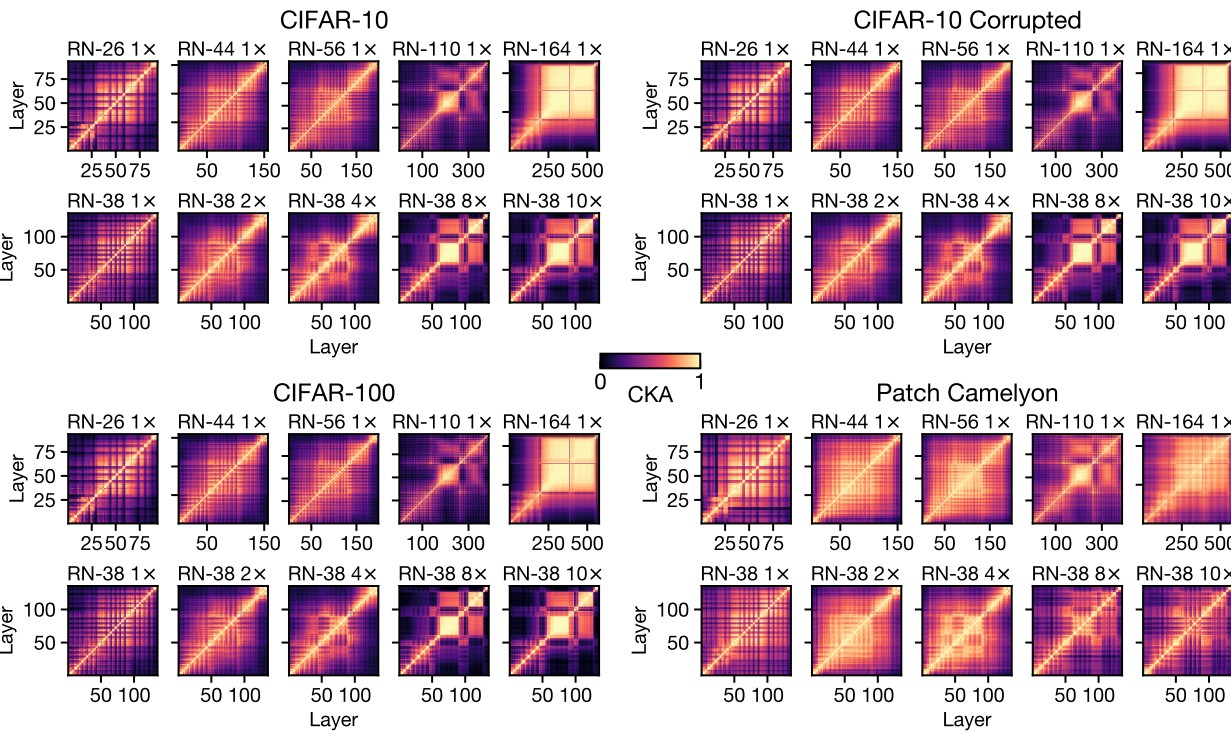

**Figure 13: Appearance of block structure depends on the data on which representations are computed.** We plot CKA heatmaps for models of varying depths (top rows) and widths (bottom rows) trained on CIFAR-10, evaluated on different datasets ordered by the degree of out-of-distribution. We observe that the block structure representation are robust to small distribution shifts in the data, as evident from CKAs computed on CIFAR-10 corrupted dataset (which adds perturbations to the original CIFAR-10 data) and CIFAR-100 dataset (which contains mutually exclusive classes but undergoes the same data collection procedure as CIFAR-10). However, larger shifts, such as from CIFAR-10 to Patch Camelyon, produce significantly different representational structures.

## C  Block Structure and the First Principal Component

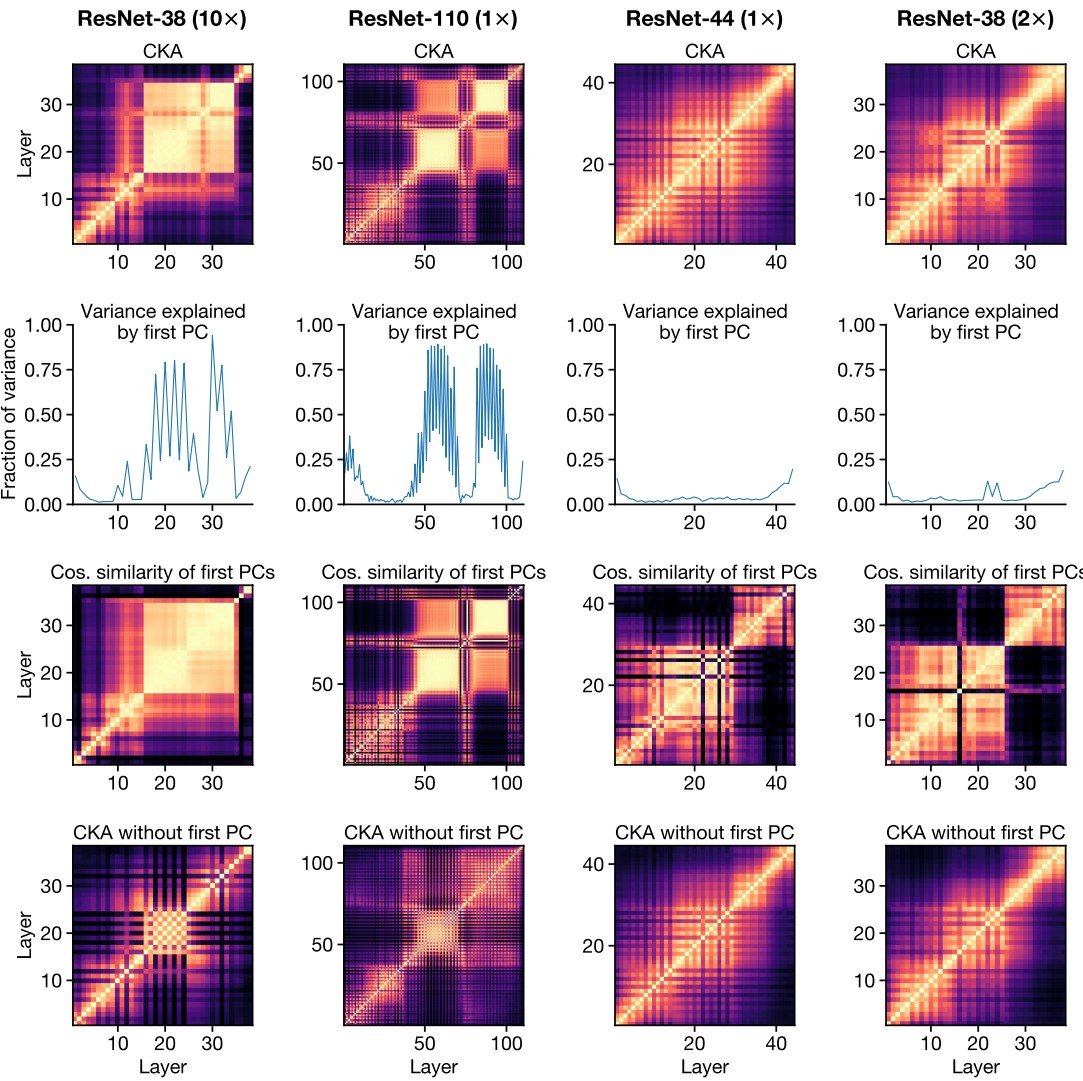

**Figure 14: The relationship between block structure and the first principal component**. Each column represents a different architecture. In ResNet-110 (1×) and ResNet-38 (10×), we observe a block structure in the CKA plot (top row), and find that the first principal component explains a large fraction of the variance in the layers that comprise the block structure (second row). We also observe that the cosine similarity of the first PCs (third row) resembles the CKA plot, and removing the first PC before computing CKA substantially attenuates the block structure (bottom row). By contrast, in ResNet-44 (1×) and ResNet-38 (2×), which have no block structure, the first PC explains only a small fraction of the variance, and the CKA plot does not resemble the cosine similarity between the first PCs, but instead resembles CKA computed without the first PCs.

# D    Additional Visualizations of Dominant Datapoints

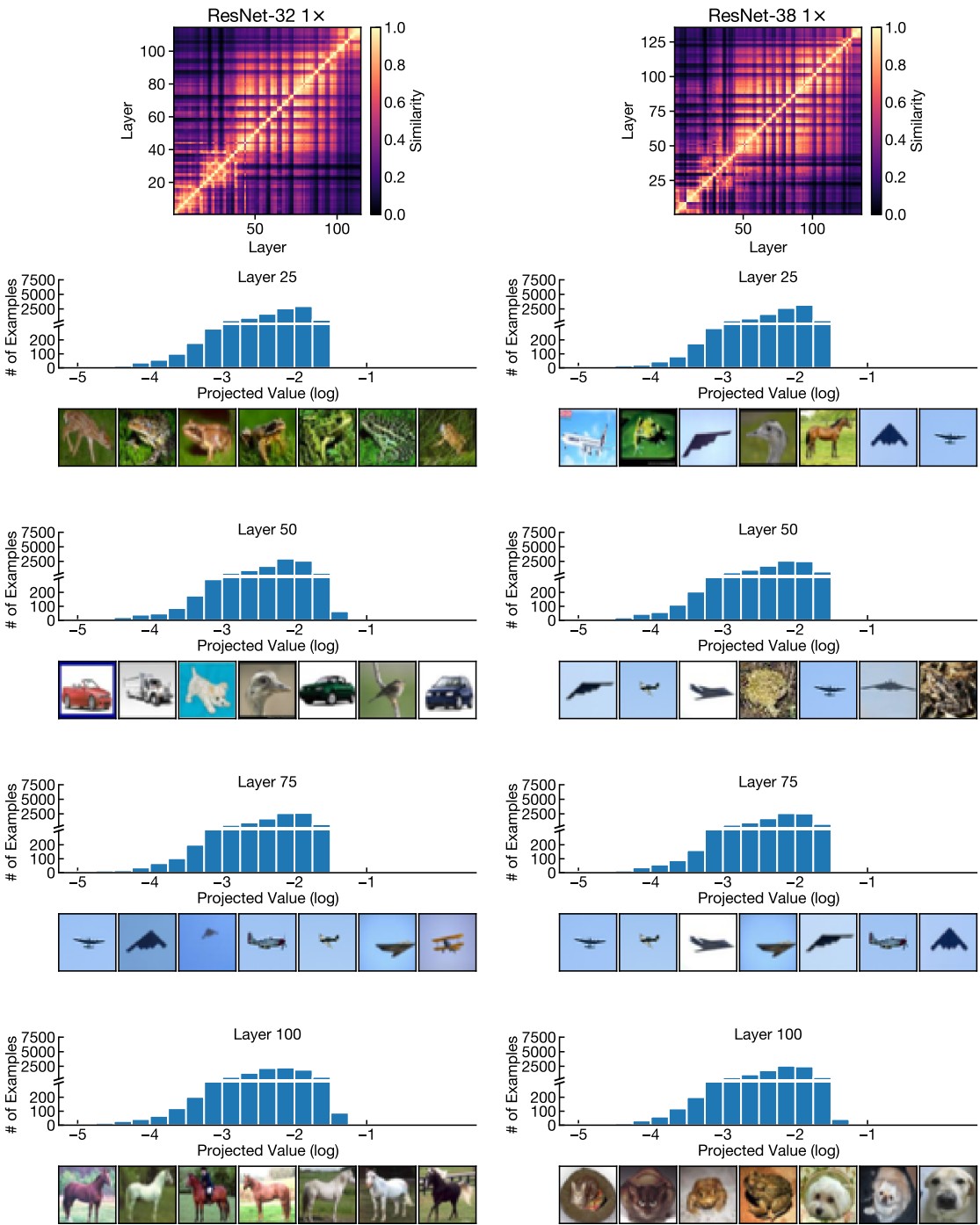

**Figure 15: Dominant datapoints are not present in networks without block structure**. The topmost row shows representational similarity heatmaps from two networks without block structure. The rows below show histograms of the projected values on the first PC, as well as images with the largest projections. Note that the distributions of projected values are unimodal, unlike the bimodal distributions observed in networks with a block structure (Figure 3). In addition, the datapoints with the highest projected values are highly dissimilar between layers.

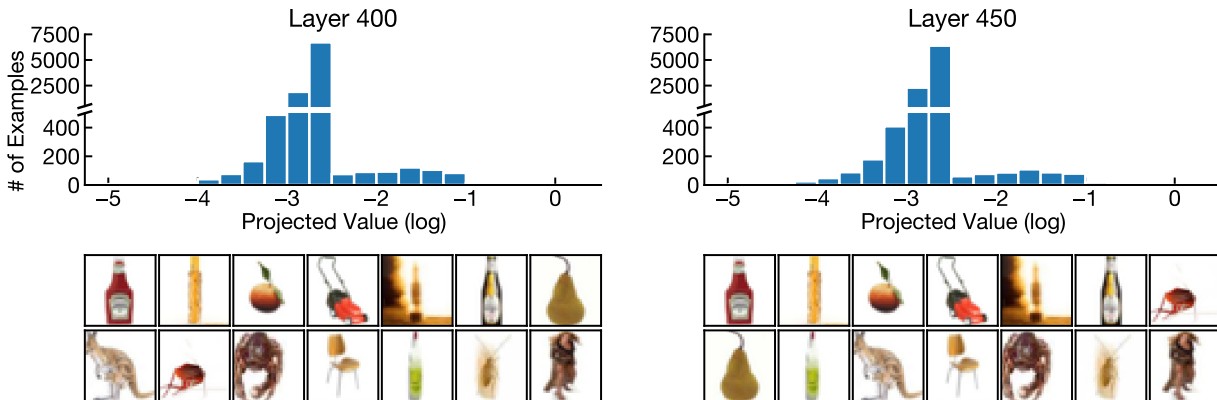

**Figure 16: Visualization of the distribution of projected values onto the first principal component by test inputs, for ResNet-224 (1×) trained on CIFAR-100.** Top row shows histograms of the projected values on the first PC. Bottom rows show images with the largest projections on the first PC. See Figure 3 for a similar plot for ResNet-164 (1×) trained on CIFAR-10.

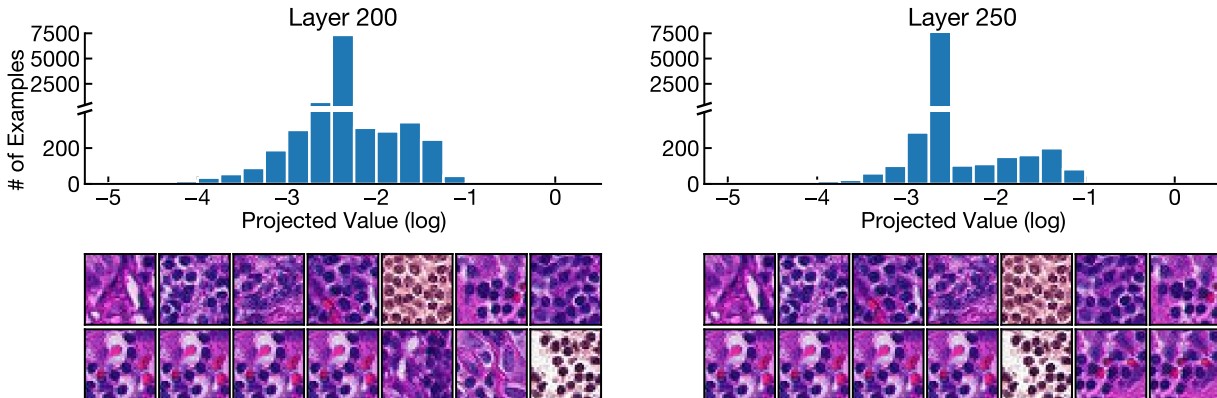

**Figure 17: Visualization of the distribution of projected values onto the first principal component by test inputs, for ResNet-80 (1×) trained on Patch Camelyon.** Top row shows histograms of the projected values on the first PC. Bottom rows show images with the largest projections on the first PC. See Figure 3 for a similar plot for ResNet-164 (1×) trained on CIFAR-10.

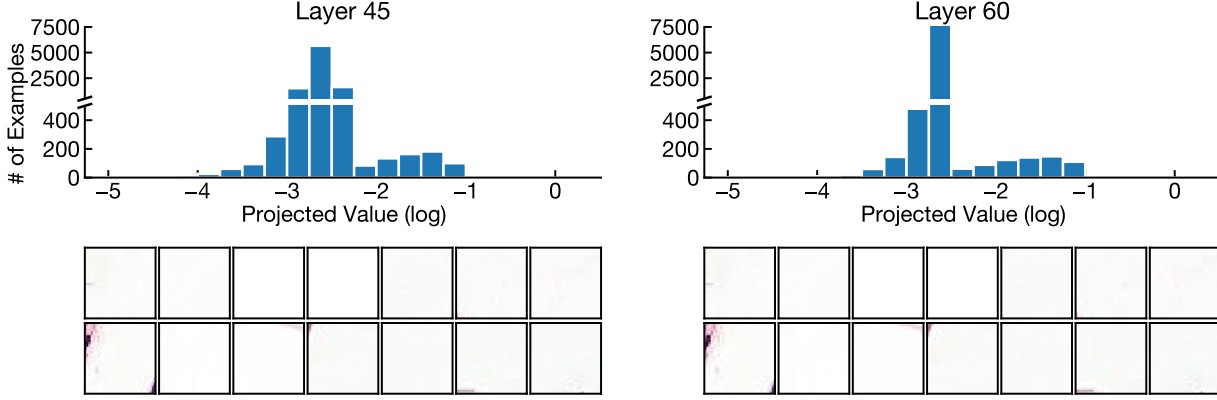

**Figure 18: Visualization of the distribution of projected values onto the first principal component by test inputs, for ResNet-26 (8×) trained on Patch Camelyon.** Top row shows histograms of the projected values on the first PC. Bottom rows show images with the largest projections on the first PC. See Figure 3 for a similar plot for ResNet-164 (1×) trained on CIFAR-10.

# E    Dominant Examples and Layer Activations

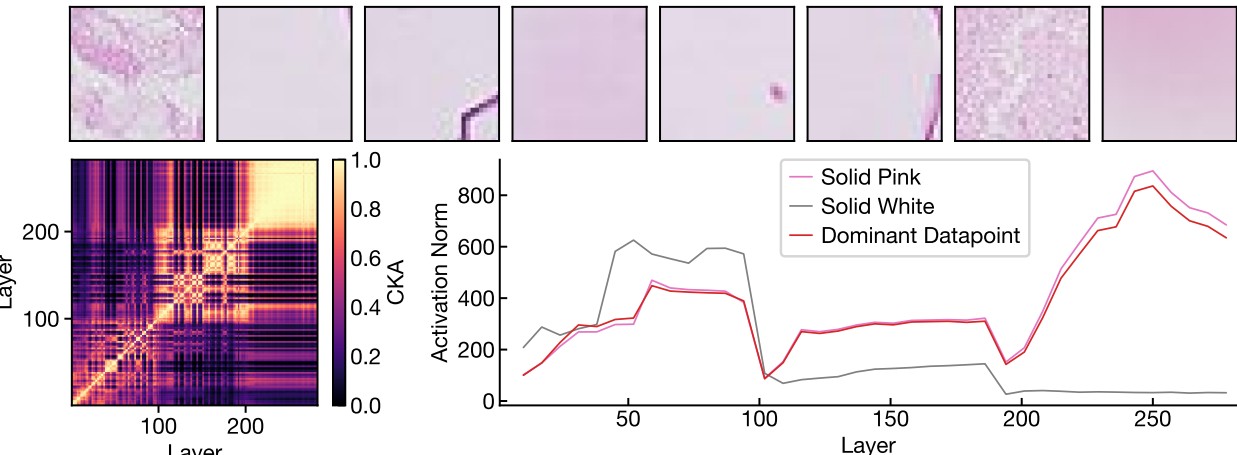

Figure 19: **Solid color images strongly activate layers making up the block structure when trained on Patch Camelyon**. Top row shows dominant examples for a ResNet-80 (1×) model trained on Patch Camelyon dataset. The CKA heatmap in the bottom left shows the location of the block structure in the internal representations of the model. We observe that the dominant images share a pink background. When we feed a synthetic image filled with this background color into the network, we observe that it yields even larger activations compared to the original image, for layers making up the block structure (i.e., after layer 200). See also Figure 7 for a similar plot for models trained on CIFAR-10 dataset.

## F    Evolution of the Block Structure

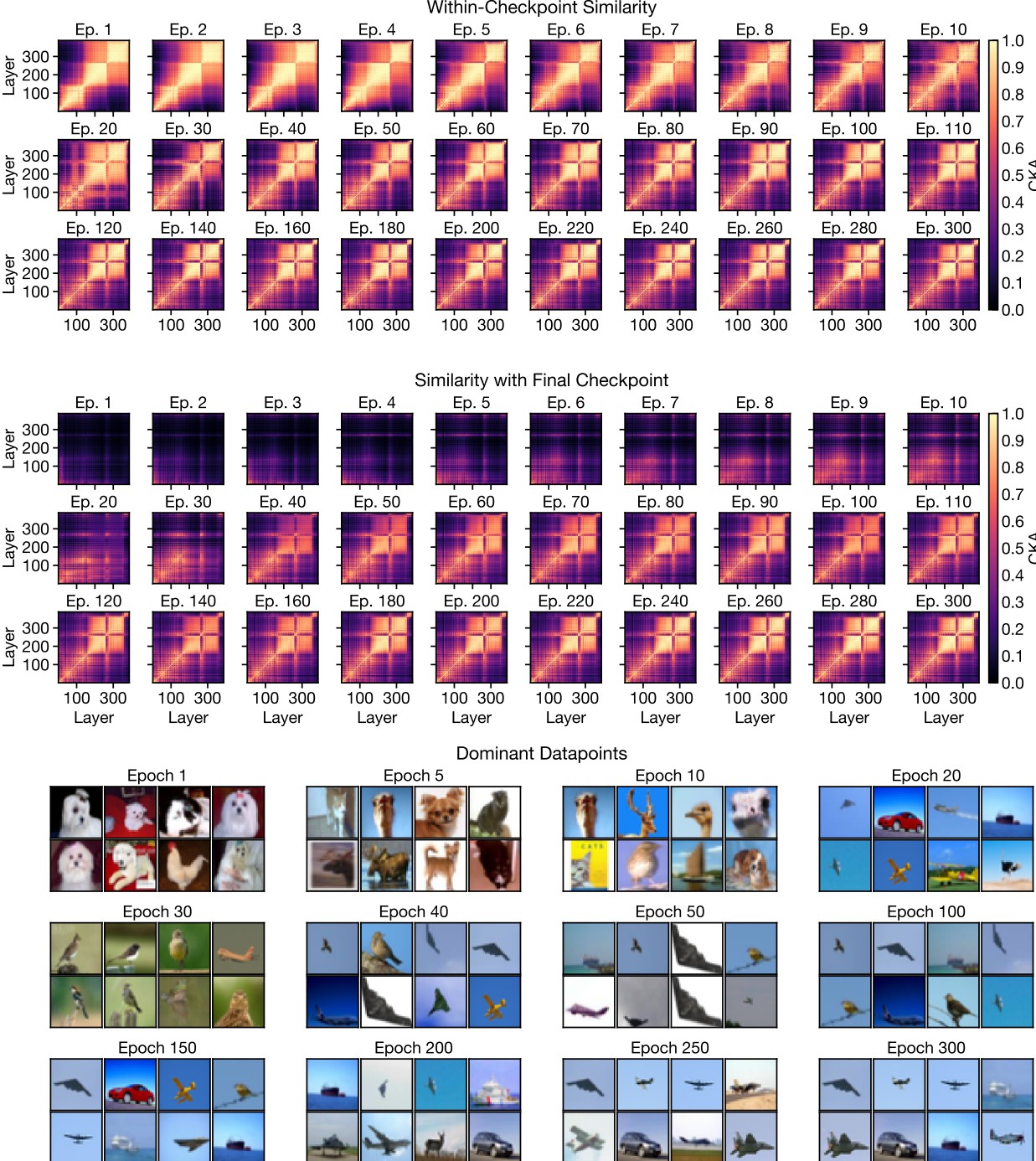

**Figure 20: Fine-grained analysis of the evolution of the block structure in a ResNet-110 (1×) model.** This plot shows the evolution of block structure for the same network as in Figure 9, but with greater temporal granularity. As in Figure 9, we find that the shape of the block structure is defined early in training (top row). However, comparing these different model checkpoints to the final, fully-trained model reveals that the block structure representations at different epochs are considerably dissimilar, especially during the first half of the training process (middle row). The corresponding dominant datapoints also vary over training, even after the block structure is clearly visible in the within-checkpoint similarity heatmaps (bottom row).

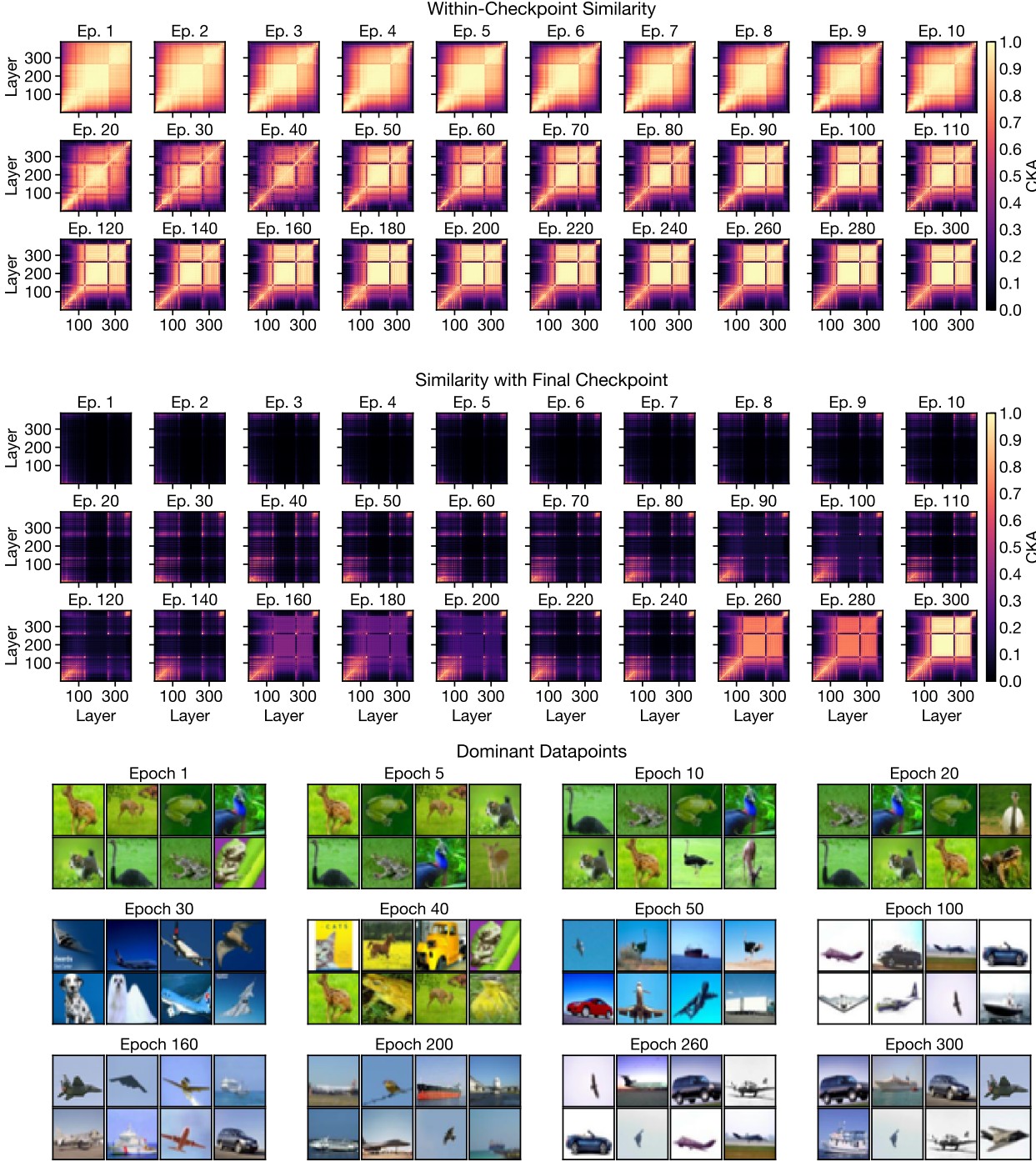

**Figure 21: Fine-grained analysis of the evolution of the block structure in a different ResNet-110 (1×) model.** This plot shows the evolution of block structure for a network that is architecturally identical to the one in Figure 9, but trained with a different seed. For this training run, the final shape of the block structure is established slightly later in training (top row), and similarity between early checkpoints and the last checkpoint is very low (middle row). Analysis of the dominant data points shows that they change substantially over the course of training (bottom row), and continue to vary long after the shape of the block structure ceases to change in the within-checkpoint similarity heatmaps.

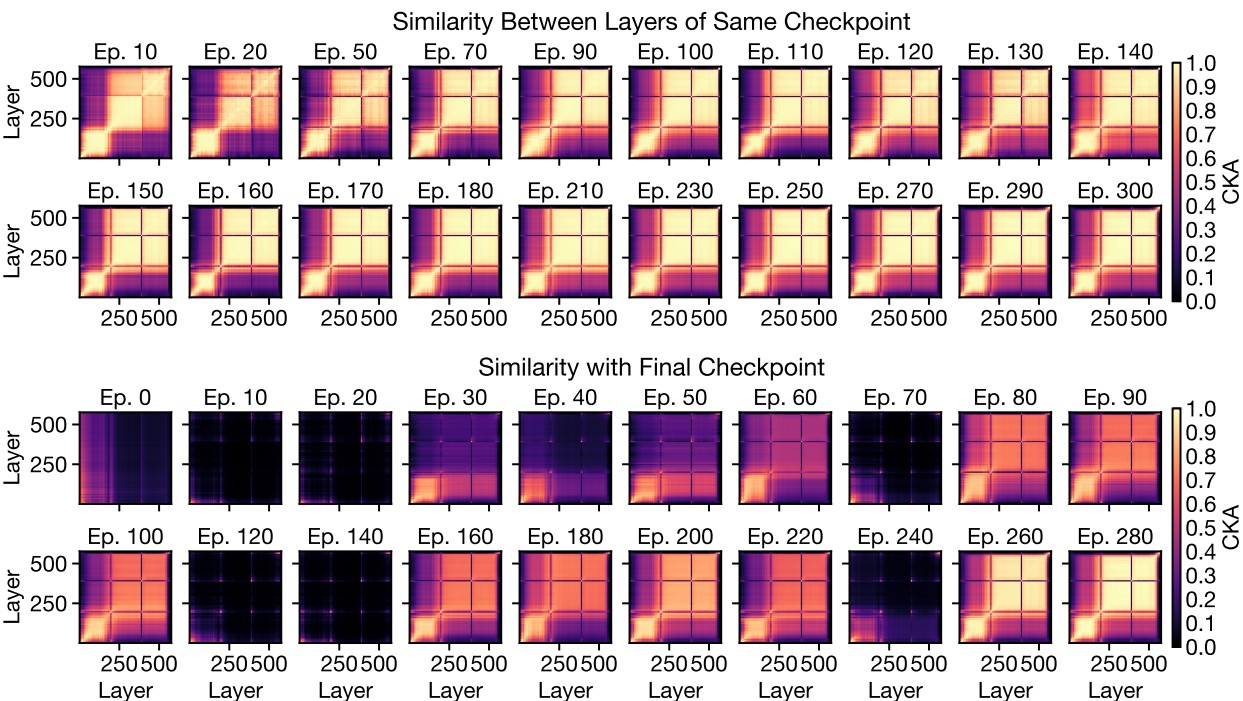

**Figure 22: Evolution of the block structure over the course of training for a ResNet-164 (1×) model**. We compute the CKA between all pairs of layers within a ResNet-164 (1×) model at different stages of training, and find that the internal representations already contain a block structure at epoch 10. Comparing these different model checkpoints to the final, fully-trained model reveals that the block structure representations at different epochs are considerably dissimilar, especially during the first half of the training process.

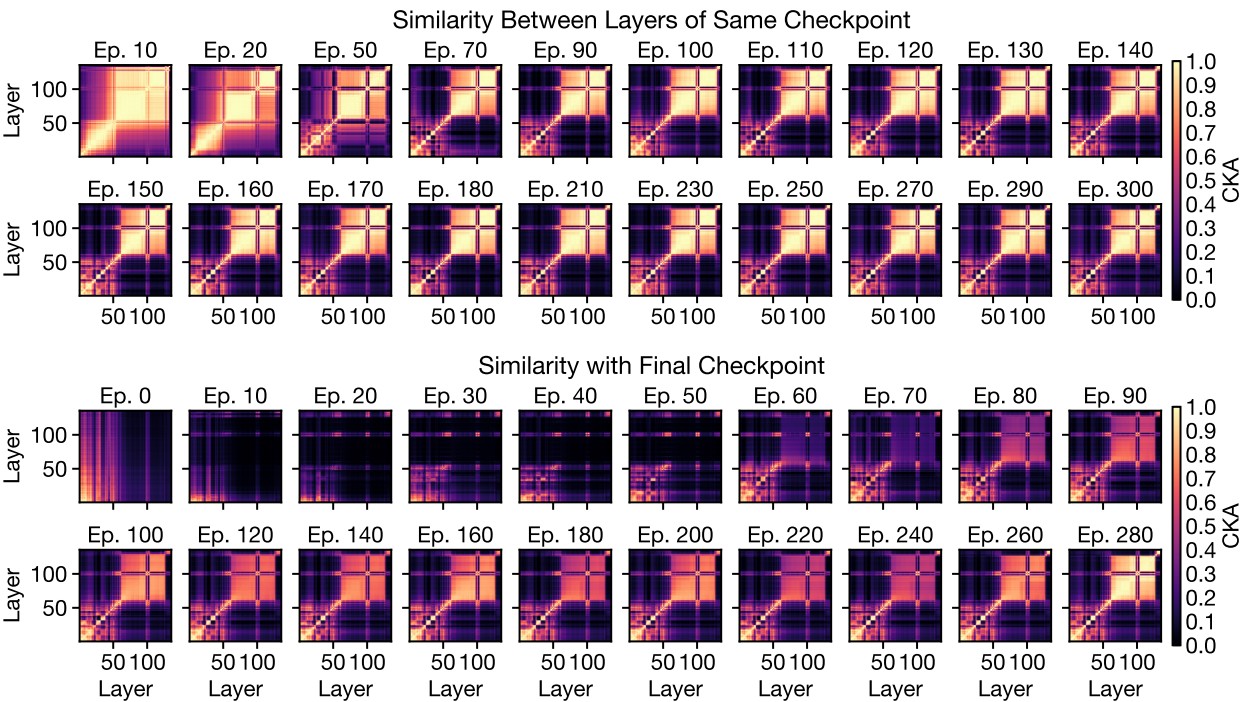

**Figure 23: Evolution of the block structure over the course of training for a ResNet-38 (10×) model.** We compute the CKA between all pairs of layers within a ResNet-38 (10×) model at different stages of training, and find that the internal representations already contain a block structure at epoch 10. Comparing these different model checkpoints to the final, fully-trained model reveals that the block structure representations at different epochs are considerably dissimilar, especially during the first half of the training process. See also Figure 22 for a similar plot for a deep ResNet (ResNet-164 (1×)).

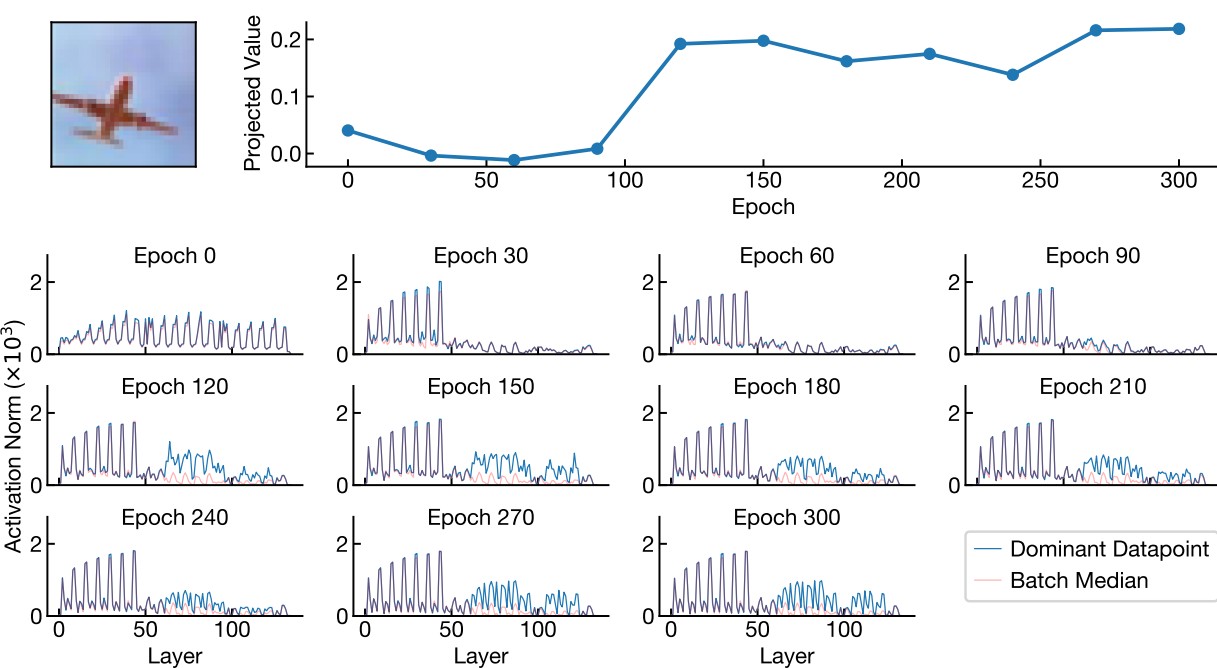

**Figure 24: Example of how the activation magnitude and the projected values onto the first principal component for a dominant image vary across different epochs, for ResNet-38 (10×).** Given a dominant datapoint for a ResNet-38 (10×) model, we track the magnitude of its projected value onto the first principal component of the block structure representations (top left), as well as its activation norms at each layer in the network (bottom set of plots), over time. We observe that before the block structure representations stabilize and show more similarity with those in the fully trained model (i.e., epoch 90, see Figure 23 above), the dominant image yields a small value when projected onto the first principal component, and also doesn't strongly activate the layers inside the block structure. This is aligned with the observation made in Figure 10 for ResNet-164 (1×).

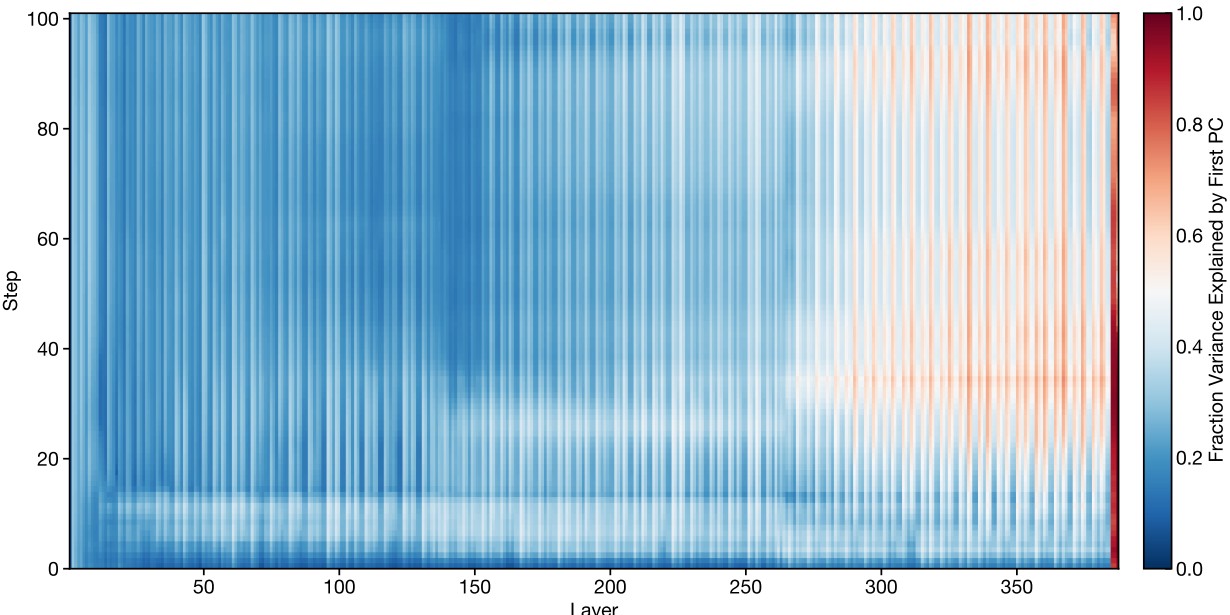

**Figure 25: Evolution of the first principal components of layer representations over the first 100 steps of training.** At every step of training, we measure the proportion of variance explained by the first principal component in each layer of a ResNet-110 (1×) network. At around step 30, the first principal component begins to explain the majority of the variance in the layer representations.

# G    Training with Principal Component Regularization

To regularize the first principal component, we first compute the amount of variance that it explains using power iteration (Miyato et al., 2018). At training step, $t$, we compute the batch of $n$ convolutional feature maps with height $h$ and width $w$ containing $c$ channels $\boldsymbol{M}_t \in \mathbb{R}^{n \times h \times w \times c}$, flatten the spatial dimensions to the channels dimension to create a matrix of size $\boldsymbol{X}_t \in \mathbb{R}^{n \times p}$ where $p = h \times w \times c$, and subtract its column means to obtain a centered matrix $\tilde{\boldsymbol{X}}_t$. We randomly initialize the stored eigenvector $\boldsymbol{u}_0 \in \mathbb{R}^p$ at the beginning of training. At each training step, we perform a single step of power iteration initialized from the previous eigenvector:

$$\boldsymbol{v}_t = \tilde{\boldsymbol{X}}_t^\mathsf{T} \tilde{\boldsymbol{X}}_t \boldsymbol{u}_{t-1} \tag{3}$$

$$\lambda_t = \|\boldsymbol{v}_t\|_2 \tag{4}$$

$$\boldsymbol{u}_t = \boldsymbol{v}_t / \lambda_t. \tag{5}$$

$\lambda_t$ approximates the top eigenvalue of $\tilde{\boldsymbol{X}}_t^\mathsf{T} \tilde{\boldsymbol{X}}_t$ and thus the amount of variance explained by the first principal component of the representation. The proportion of variance explained is given by $\lambda_t / \|\tilde{X}_t\|_\mathrm{F}^2$. We incorporate the regularizer as an additive term in the loss:

$$\mathcal{L}_{\mathrm{pc\_reg}}(\lambda_t, \tilde{\boldsymbol{X}}; \alpha, \delta) = \alpha \max(\lambda_t / \|\tilde{\boldsymbol{X}}_t\|_\mathrm{F}^2 - \delta, 0), \tag{6}$$

where $\alpha$ is the strength of the regularizer and $\delta$ is the threshold proportion of variance explained at which it is imposed. In our experiments, we tune $\alpha$ as a hyperparameter with values in $[0.1, 1, 10]$, and set $\delta = 0.2$ based on our analysis of the first principal components of models without the block structure. To speed up the training process, we only apply the regularizer to ReLU layers starting from the second stage, where block structure is often found.

# H   Effect of Batch Size on the Block Structure

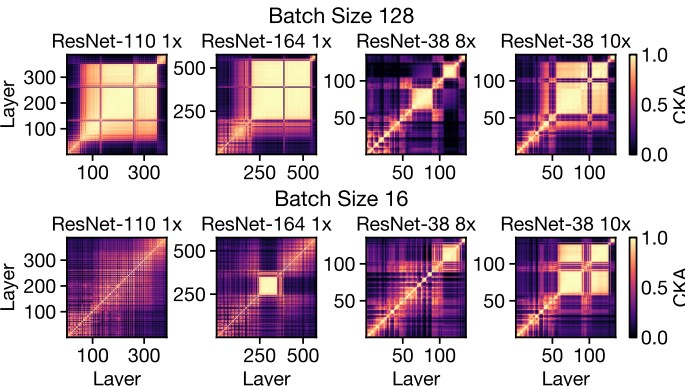

**Figure 26: Using very small batch sizes during training reduces the appearance of the block structure**. The top row shows the block structure effect in a range of very deep and wide networks, trained with standard batch size = 128. We experiment with a drastically smaller batch size of 16 (bottom row) and find that the block structure is now highly reduced, especially in deep models.

# I   Impact of Transfer Learning and Shake-Shake Regularization on Similarity of Layers Inside the Block Structure

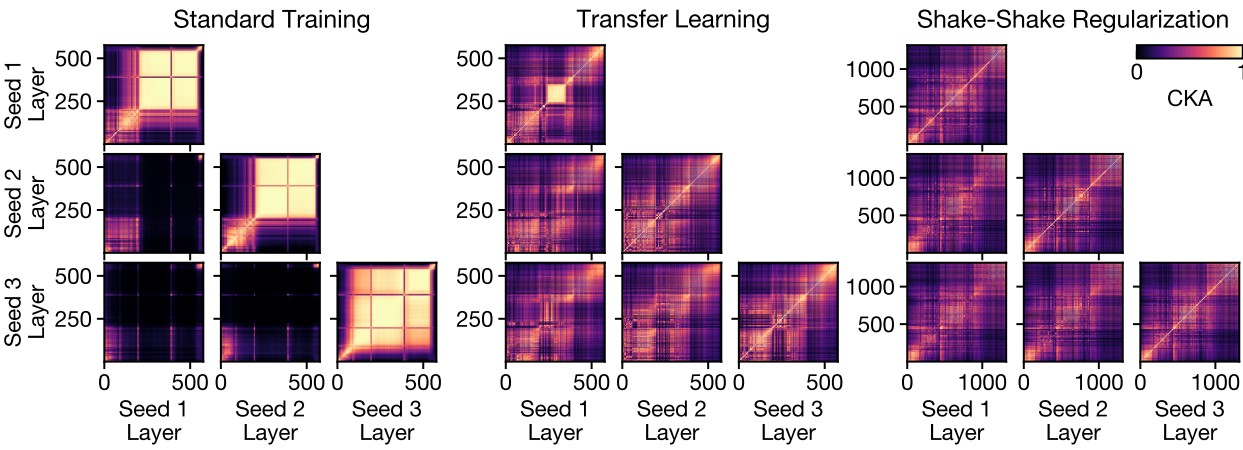

**Figure 27: Training with transfer learning and Shake-Shake regularization yields models that are more similar across different training runs**. Each group of plots shows CKA between layers of models with the same architecture but different initializations (off the diagonal) or within a single model (on the diagonal). In the standard training case, representations across models are highly dissimilar, especially in the block structure region. In contrast, when we use transfer learning and Shake-Shake regularization, comparisons across seeds show more similarity in corresponding layers.

## J Block Structure Under Different Kernels

As previously identified by (Nguyen et al., 2021), the block structure is a phenomenon the linear CKA heatmaps of large (wide or deep) networks. In this section, we investigate whether the block structure phenomenon also arises in CKA heatmaps computed with other kernels, and also examine the effect of removing the dominant datapoints (identified by the magnitudes of their projections on the first principal component, as in Section 4.1) upon these CKA heatmaps.

To compute CKA heatmaps under alternative kernels, we again use minibatch CKA. The approach in Eq. 1 can be easily adapted to nonlinear kernels by replacing $\boldsymbol{X}_i \boldsymbol{X}_i^\mathsf{T}$ and $\boldsymbol{Y}_i \boldsymbol{Y}_i^\mathsf{T}$ the linear Gram matrices formed by minibatch $i$, with minibatch kernel matrices $\boldsymbol{K}_i \in \mathbb{R}^{n \times n}$ and $\boldsymbol{K}_i' \in \mathbb{R}^{n \times n}$. The elements of these minibatch kernel matrices are the kernels between pairs of examples in the minibatches, i.e., $K_{i_{lm}} = k(\boldsymbol{X}_{i_{l,:}}, \boldsymbol{X}_{i_{m,:}})$ and $K_{lm}' = k'(\boldsymbol{Y}_{i_{l,:}}, \boldsymbol{Y}_{i_{m,:}})$. Like linear minibatch CKA, nonlinear minibatch CKA is computed by averaging $\mathrm{HSIC}_1$ across minibatches:

$$\mathrm{CKA}_{\mathrm{minibatch}} = \frac{\frac{1}{k}\sum_{i=1}^k \mathrm{HSIC}_1(\boldsymbol{K}_i, \boldsymbol{K}_i')}{\sqrt{\frac{1}{k}\sum_{i=1}^k \mathrm{HSIC}_1(\boldsymbol{K}_i, \boldsymbol{K}_i)}\sqrt{\frac{1}{k}\sum_{i=1}^k \mathrm{HSIC}_1(\boldsymbol{K}_i', \boldsymbol{K}_i')}}. \tag{7}$$

We investigate the behavior of CKA under the linear kernel $k_{\mathrm{linear}}(\boldsymbol{x}, \boldsymbol{y}) = \boldsymbol{x}^\mathsf{T}\boldsymbol{y}$, the cosine kernel $k_{\cos}(\boldsymbol{x}, \boldsymbol{y}) = \boldsymbol{x}^\mathsf{T}\boldsymbol{y}/(\|\boldsymbol{x}\|\|\boldsymbol{y}\|)$, and the RBF kernel $k_{\mathrm{rbf}}(\boldsymbol{x}, \boldsymbol{y}; \sigma) = \exp(-\|\boldsymbol{x} - \boldsymbol{y}\|^2/(2\sigma^2))$. For each layer, we measure the median Euclidean distance $\tilde{d}$ between examples in each layer and set $\sigma = c\tilde{d}$ with $c \in \{0.2, 0.5, 1, 2, 5, 10\}$ of that median Euclidean distance. To reduce variance when computing RBF CKA with small $c$, we use a minibatch size of 1000 for these experiments.

Figure 28 shows the appearance of CKA heatmaps of a narrow, shallow network (ResNet-38 1×, top), a wide network (ResNet-38 10×, middle), and a deep network (ResNet-164 1×, bottom). Although heatmaps computed for a small network (ResNet-38 1×) look qualitatively similar regardless of kernels, both wide (ResNet-38 10×) and deep (ResNet-164 1×) networks exhibit significant differences.

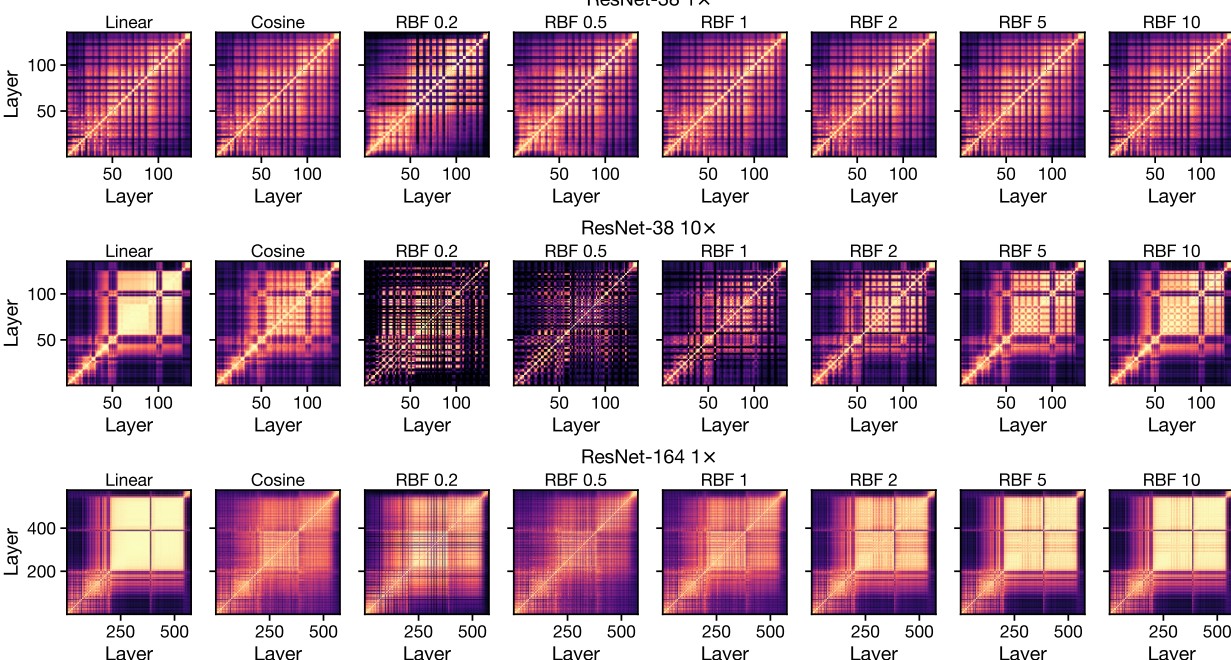

**Figure 28: Appearance of representation heatmaps in wide and deep networks, but not narrow/shallow networks, depends on the choice of kernel**. Rows reflect different models and columns reflect different kernels. For RBF kernels, the parameter indicates the fraction of the median distance between examples (computed separately for each layer) that is used as the standard deviation of the kernel.

Because differences in representational similarity heatmaps ultimately reflect differences in the underlying kernel matrices, in Figure 29, we show kernel matrices of individual layers taken from inside the block structure of each network on random minibatches where the examples have been sorted in descending values of the first principal component. All kernels are sensitive to dominant datapoints, but in different ways and to different degrees. Linear kernel matrices are dominated by the similarity between dominant datapoints. The cosine kernel ignores activation norms, and finds high similarity within groups of dominant and non-dominant datapoints but low similarity between groups. The RBF kernel effectively considers all far away points to be equally dissimilar, and thus indicates that dominant datapoints are dissimilar to all other datapoints, including other dominant datapoints, which are typically far in Euclidean distance (because, while aligned in direction, they have different norms).

Note that the prevalence of dominant datapoints can differ across models and initializations, as previously demonstrated in Figure 5. The dominant datapoints are clearly visible as a block in the top-left corner of the cosine kernel matrix. For ResNet-38 10×, there are 14 in the minibatch of 128 examples that is shown, but for ResNet-164 (1×), there are only 2.

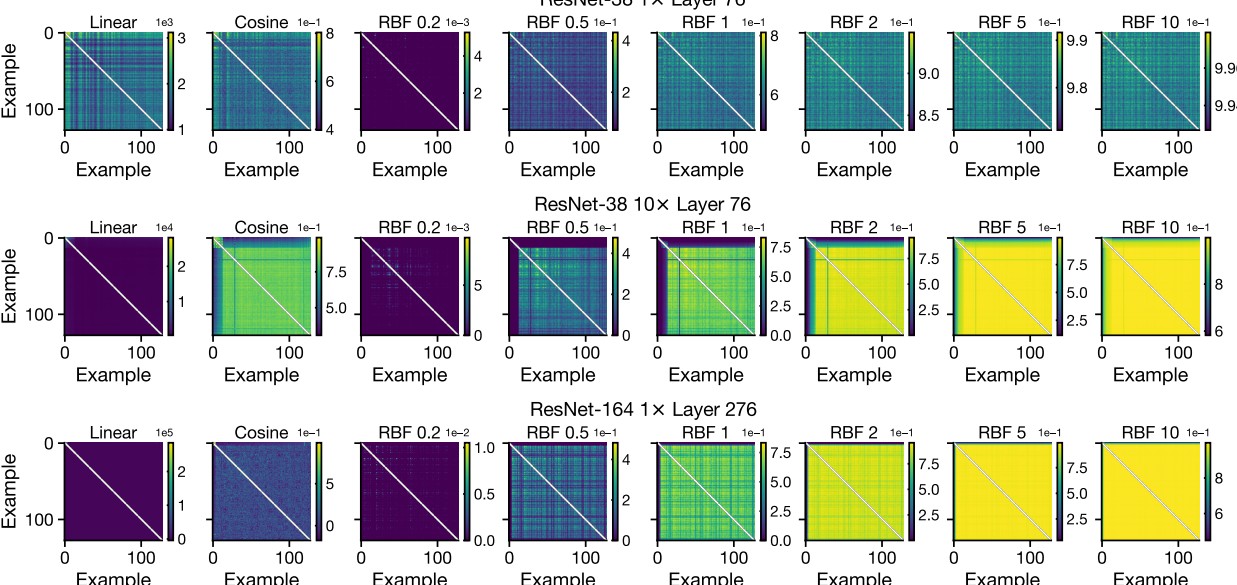

**Figure 29: Kernels based on dot products, cosine similarity, or Euclidean distance are sensitive to dominant datapoints**. Plots show kernel matrices for a randomly sampled minibatch of 128 examples, computed from a layer inside the block structure, for models that exhibit one. Examples are sorted in descending value of the first principal component of the raw activations; top rows and left columns reflect dominant datapoints.

What is the effect of removing dominant datapoints upon CKA similarity heatmaps computed with these other kernels? In Figure 30, we show that, once the dominant datapoints are removed from large networks, we again see only differences among CKA heatmaps computed with different kernels, in line with the results observed for shallow networks in Figure 28. There are no longer large blocks of many consecutive similar layers in any of the heatmaps. Across all choices of kernel that we have investigated, when blocks appear in CKA heatmaps, they can be eliminated by eliminating the dominant datapoints.

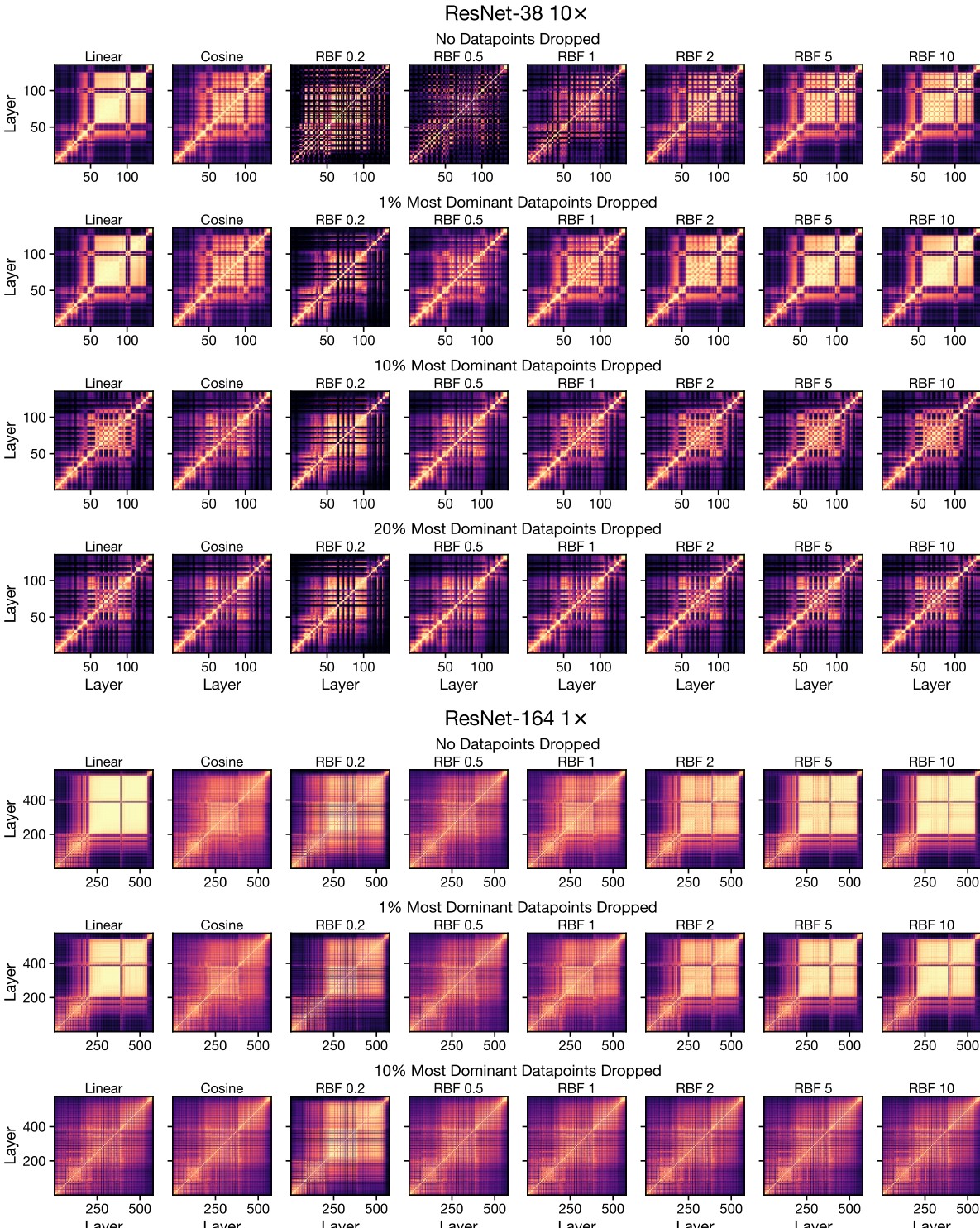

**Figure 30: Removing dominant datapoints eliminates both blocks and differences among kernels in CKA representational similarity heatmaps**. We remove the top k% of examples according to the magnitudes of their projections on the first PC in layer 76 for ResNet-38 10× and layer 276 for ResNet-164 1×. Because 14/128 datapoints are dominant in the minibatch kernel matrix shown in Figure 29, we provide results for removing 20% of datapoints for that network, although they look only modestly different from the results obtained by removing 10% of datapoints.

