# OpenReview forum: "On the Origins of the Block Structure Phenomenon in Neural Network Representations"
_TMLR — Accepted by TMLR_

### Review · Reviewer_GD8Z · 2022-08-28

**Summary Of Contributions:**

The paper provides an improved understanding on the phenomenon of block structures ,i.e. high levels of similarity among consecutive neural network layers, arising within the intermediate representations of deep neural networks. In particular, it reconciles the seemingly contradictory observations that block structures regularly occur (suggesting good fitting to the data and generalization) while the resulting block representations being quite variable across training seeds (suggesting overfitting to noise in the data). Indeed, the paper studies a variety of deep neural networks and proposes an explanation based on *dominant data points*. Dominant data points, which themselves are a small proportion of the overall data provided in training, ultimately yield strong activations at the beginning of training, leading to the block structures empirically observed. These data points typically share several high-level features, e.g., background colors, and are dictated by the statistics of the input dataset. However, these data points are not consistent throughout training, and in fact change significantly, both during training and across different training initializations due to batch and initialization randomness, which in turn explains the variability in block structures.

To validate the influence of dominant data points on block structures, multiple experimental protocols are proposed. First, dominant data points are shown to yield large activations within neural networks, thus dictating the principal components of their intermediate representations. This is further confirmed through the removal of said data points, following which block structures disappear. The underlying shared properties of these data points are also explored through an augmentation experiment, in which solid color images, based on dominant data points, are introduced, and these in turn lead to even higher activations than the original images. Moreover, the variability of the block structure is analyzed, and shown to change significantly over the course of training: Dominant data points are not static, but rather change substantially over time, likely due to the minibatch training process and to random initial model configurations. Finally, the paper studies regularization techniques, principal component and shake-shake regularisation, showing that they can effectively alleviate block structures without harming model performance (even at times yielding minor improvements), and mentions that transfer learning can also help mitigate this phenomenon.

**Broader Impact Concerns:**

No broader impact concerns.

**Requested Changes:**

1) Please provide a more in-depth analysis, ideally supported by probabilistic/theoretical arguments, for the properties of dominant data points. It is currently clear that data points exhibit statistically meaningful high-level features, yet this is a somewhat hand-wavy insight that really should be studied in more depth. Moreover, this becomes more confusing when considering that random seeds and training significantly affect which data points become dominant. Hence, I strongly recommend that such an analysis be conducted, so as to understand jhow dominant data points are brought about, both from a data and model perspective.

2) On a more minor note, I believe that the manuscript should be revised to minimise references to the appendix for essential arguments that are key to the flow of the paper.

**Strengths And Weaknesses:**

**Strengths:**

The paper provides a very comprehensive empirical analysis of the block structure phenomenon, and conducts an extensive battery of experiments to support its claims. The experiments conducted towards validating hypothesis are sound, well-thought, and convincing. In particular, the data augmentation experiments highlight the impact of dominant data points, and the visualisations of activation distributions, prominent data points, and layer-wise similarity heatmaps clearly highlight the role of different data points during model training, as well as the significant change in these data points over time. From a writing perspective, the paper is easy to follow and is very well illustrated. The contributions are accurately stated at the beginning of the manuscript and the main findings are discussed well.

**Weaknesses:**

Though the empirical analysis of this phenomenon is quite extensive and concrete, I feel that the paper lacks a theoretical component/discussion supporting its claims. For instance, the intuitions about dominant data points seem well-justified, and the experimental results showing their effects appear solid. However, it is not clear *why* this phenomenon arises from reading the paper, or what sets of properties make a data point dominant. The paper mentions that dominant data points present high-level features that are statistically meaningful, e.g., background color, but do not investigate this point any further. Moreover, the discussion on variability of dominant data points also necessitates explanation. More specifically, it would be very interesting to explore what conditions enable data points to arise as dominant near the beginning of training, as this could offer an avenue to solving the problem beyond the more patch-like regularisation techniques discussed. Indeed, this ties in nicely with the earlier point on "What makes a data point dominant?" Clearly, the answer lies both within the dataset statistics and in the randomness of the ensuing model/training, but this line of thought is not expanded on, and this severely limits the applicability and relevance of the paper's insights more broadly.

From a writing perspective, I find that the paper would benefit strongly from re-positioning some of its sections and including many sections from the appendix. Throughout the main body, the appendix is  heavily referenced, and as a result the paper cannot be read in a stand-alone fashion. I therefore recommend that the authors revisit their writing so as to make the main body self-contained.

---

> ### Author Response · Authors · 2022-09-22
> **Response to Reviewer GD8Z**
>
> **Lack of a theoretical component.** We respectfully disagree with the need for a theoretical discussion of our findings. As the reviewer recognizes, the phenomenon we study is complex—it depends on the dataset statistics, architecture, and seed, and dominant data points do not appear until several dozen steps into training. Given this complexity, we have so far been unable to find a theoretical approach that yields insight into why it occurs. However, the existence of the phenomenon does not depend on whether we have a theory for it. Empirical studies that aim to understand properties of neural networks are not uncommon—a notable example of this line of work includes research on Lottery Ticket Hypothesis [1]. Overall, given that we make no theoretical claims, we do not believe that the absence of theory should be used as a reason to reject our paper.
>
> **Minimize references to the appendix.** Per your comment, we have moved some content from the appendix to the main text. This includes Figures 4, 10 and 11 as well as Table 1. The remaining figures in the Appendix simply demonstrate additional results on other datasets/architectures beyond those that are already discussed in the main text.
>
> **References**
>
> [1] Frankle, Jonathan, and Michael Carbin. "The lottery ticket hypothesis: Finding sparse, trainable neural networks." arXiv preprint arXiv:1803.03635 (2018).

---

### Review · Reviewer_LF8A · 2022-08-29

**Summary Of Contributions:**

This paper looks at the block structure phenomenon which arises during standard training of high capacity NNs.  They find that this phenomenon arises from what they introduce as dominant datapoints, which are a small portion of datapoints which have significantly larger value when their activation is projected onto the first principle component.  They find that these dominant datapoints often share similar backgrounds.  The authors also look at the evolution of block structure during training and find that the block structure arises early during training, but the set of dominant datapoints varies throughout training.  Additionally, they look at the impact of various forms of regularization on block structure and find that regularization can remove block structure without hurting accuracy.

**Requested Changes:**

- clarifying the motivation of the study of the block structure phenomenon (would strengthen work in my view)

**Strengths And Weaknesses:**

Strengths:
- scope of experiments- I appreciate the scope of experiments in the paper.  The authors experiment with multiple datasets and architectures and find similar patterns across datasets/architectures.  I also liked the design of experiments in the paper since they did a great job of demonstrating that the block structure phenomenon originates from dominant datapoints.  For example, the authors include experiments where the remove a fraction of the dominant datapoints from evaluation and find that the block structure disappears and show that for models without block structure, there are no dominant datapoints.
- writing is clear

Weaknesses:
- motivation- while the results are interesting, the motivation for the paper is not very clear to me.  Why should we be interested in the block structure phenomenon?  Can knowledge of the origin of the block structure phenomenon guide us towards improving training methods for high capacity NNs?  While the paper does look at different regularization techniques and their impact on block structure and training accuracy, it isn't very clear to me whether we should be trying to learn models that avoid block structure or not.

---

> ### Author Response · Authors · 2022-09-22
> **Response to Reviewer LF8A**
>
> **Clarifying the motivation of the paper.** The main focus of our paper is to resolve the discrepancies/missing pieces from previous work by Nguyen et al., 2021 [7]. It’s not our goal to explicitly improve performance, even in Section 6 where we investigated the interplay between training interventions and block structure representations. Given that the block structure phenomenon is prevalent in many overparameterized networks, our work seeks to rigorously characterize its cause and effects, and consequently provide a better understanding of the nuances in the way these networks learn, despite their similarly good performances.
>
> Based on your feedback, we have rewritten parts of the Introduction to further clarify the motivation of our investigation.
>
> **Why should we be interested in the block structure phenomenon?** One way to understand how neural systems work is to examine what information is represented in these intermediate representations and how that information is transformed from one layer to the next. We believe that this form of understanding is meaningful independent of any connection to other functional/computational properties, and our paper fits into previous literature that follows a similar approach (e.g. [1,2,3,4]). We also note that this representational approach to understanding neural systems is the dominant paradigm in modern neuroscience systems (e.g. [5,6]).
>
> The block structure phenomenon, in particular, explains most of the variance in the hidden representations of many layers in high-capacity networks, but it is not present at all in low-capacity networks. Thus, it reflects the single largest representational change associated with overparameterization. Rigorously characterizing representational properties of this phenomenon is a prerequisite for drawing any kind of connection to other functional properties of neural networks. Moreover, these representational properties have implications for techniques that make more direct use of internal representations, such as transfer learning and some interpretability methods.
>
> **References**
>
> [1] Raghu, A., et al. (2019, September). Rapid Learning or Feature Reuse? Towards Understanding the Effectiveness of MAML. In International Conference on Learning Representations.
>
> [2] ​​Cohen, U., Chung, S., Lee, D. D., & Sompolinsky, H. (2020). Separability and geometry of object manifolds in deep neural networks. Nature Communications, 11(1), 1-13.
>
> [3] Merel, J., et al. Deep neuroethology of a virtual rodent. In International Conference on Learning Representations.
>
> [4] Gotmare, Akhilesh, et al. "A closer look at deep learning heuristics: Learning rate restarts, warmup and distillation." arXiv preprint arXiv:1810.13243 (2018).
>
> [5] Kriegeskorte, N., et al. (2008). Representational similarity analysis-connecting the branches of systems neuroscience. Frontiers in systems neuroscience.
>
> [6] Stringer, C., et al. (2019). High-dimensional geometry of population responses in visual cortex. Nature, 571(7765), 361-365.
>
> [7] Nguyen, T., et al. "Do wide and deep networks learn the same things? uncovering how neural network representations vary with width and depth." International Conference on Learning Representations (2021).

---

### Review · Reviewer_uytN · 2022-09-08

**Summary Of Contributions:**

The paper analysis the block structure observed in neural networks. It identifies connections to the dataset statistics, training procedure and the network size. Further, it gives some suggestions on how to reduce these structures. The behaviour has been analyzed for several network architectures and datasets.

**Requested Changes:**

Important
* Adding the effect of the described methods in section 6 on the accuracy, otherwise the made claim that eliminating the block structure is not interfering with generalization is not fully evaluated.

Changes which would strengthen the work
* Making the motivation clearer on why eliminating the blockstructure is necessary


**Strengths And Weaknesses:**

**Strengths**
* Well written and clear
* Analysis of the origins of the block-based structure
* Has some interesting insights (connection of the structure with model complexity, the statistics of the training dataset, used augmentations, training procedure...)

**Weaknesses**
* One of the claimed contributions is how to eliminate these block structure. However, whether this actually is necessary remains unclear from the introduction.
* Further, the effect of the proposed interventions on the accuracy of a network is only presented for one of the described methods and only in the Appendix. In my opinion this should be part of the main paper as well. Otherwise it is unclear, whether it is actually necessary to remove/avoid these block structures.

---

> ### Author Response · Authors · 2022-09-22
> **Response to Reviewer uytN**
>
> **Adding the effect of the described methods in section 6 on the accuracy.** Per your comment, we have moved Table 1 from the Appendix to Section 6 of the main text. We also added a new table for comparing standard training performance on CIFAR-10 with the accuracies obtained from other training interventions (i.e., Shake-shake regularization and transfer learning).
>
> **Making the motivation clearer on why eliminating the block structure is necessary.** We do not believe that it is necessary to remove block structure from the internal representations. Our goal in these experiments is simply to investigate the impact of doing so. We find that the impact upon accuracy is positive but small, suggesting that the effect of the block structure on generalization is minor. We have revised the introduction of the paper as well as Section 6 to clarify our interpretation of these results.

---

### Review · Reviewer_JN8m · 2022-09-09

**Summary Of Contributions:**

The paper empirically studies the similarities and correlations between network responses at different layers, called the block-structure effect after the work by Nguyen et al., 2021, that emerge in deep neural networks after training them on a classification dataset. The primary finding of the paper is that only a small subset of the training datasets, called dominant data points, contribute to this phenomenon. It is observed that these data points share high-level semantic similarities, e.g., having similar backgrounds. It is hypothesized that since according to empirical observations, these data points change depending on the random seed, the exact block-structure effect will also be different for different initialization random seeds. Experiments are performed by training ResNet-based architectures using CIFAR10, CIFAR100, and Patch Camelyon datasets.

**Broader Impact Concerns:**

No concern could be detected.

**Requested Changes:**

I think this paper is not ready for publication and needs significant improvement:

1. I agree some of the observations are non-trivial but the motivation is unclear. It is unclear to me what is the goal of the explorations in this work. What will be the benefits of the findings for the ML community? I think the the introduction portion of the paper needs to be rewritten. It is not clear to me why the block structure is even important, to begin with. The motivation, importance, and benefits of this study must be stated clearly.

2. Conclusions are based on empirical investigations that are very limited in my opinion. It is not easy to draw conclusions by running experiments on one network and three datasets. At least several architectures should be considered and more complex datasets need to be used in experiments to draw more confident conclusions. It is really non-trivial to me that the findings of experiments are generalizable.

3. The paper lacks theoretical insight. I understand the field of deep learning has a highly empirical nature but when a work is about providing insights into deep neural networks, I expect more justifications.

In conclusion, this work needs significant improvements to become suitable for publication.

**Strengths And Weaknesses:**

Strengths:

1. Experiments provide non-trivial observations that may be interesting.

2. The paper is technically sound.


Weaknesses:

1. The motivation and the goal of the proposed study are unclear.

2. Conclusions are based on experiments with a relatively limited scope.

---

> ### Author Response · Authors · 2022-09-22
> **Response to Reviewer JN8m**
>
> **What will be the benefits of the findings for the ML community?...The motivation, importance, and benefits of this study must be stated clearly.** The TMLR reviewer guidelines state that significance and impact should not be a factor in acceptance decisions at TMLR as long as a paper is interesting to “at least some individuals in TMLR's audience,” and if the reviewer is uncertain whether the paper satisfies this criterion, they should “assume that it does.” In our case, the goal of our work is to resolve some discrepancies/missing pieces from previous work by Nguyen et al., 2021 [1]. We believe that our submission will be of interest to readers of that paper and should thus easily satisfy the TMLR review criteria.
>
> However, we accept that our previous introduction had room for improvement, and we have revised it to clarify why we chose to pursue this direction of research. To briefly restate, the block structure phenomenon corresponds to the largest representational change that we observe as we make our ResNets wider/deeper. We find it surprising that there exists a large range of layers with highly similar representations. We felt (and still feel) that a better understanding of how the representations were actually changing with overparameterization is valuable.
>
> **Conclusions are based on empirical investigations that are very limited.** We respectfully disagree with this point. In terms of model, we experimented with the ResNet backbone commonly used for CIFAR and the one commonly used for ImageNet. Within each model family, we covered a substantial range of model depths and widths. In terms of dataset, besides CIFAR-10 and CIFAR-100, we have also demonstrated the presence of dominant datapoints in models trained on ImageNet, as well as Patch Camelyon, a medical imaging dataset. We believe that this is a sufficient scope for an empirical investigation.
>
> **The paper lacks theoretical insight.** We respectfully disagree with the need for a theoretical discussion of our findings. The characteristics of the empirical phenomenon we study make it difficult to characterize this phenomenon using the standard tools of deep learning theory—it is not present at initialization, and it depends on both the network width/depth and properties of the dataset—but the empirical phenomenon exists whether or not there is a theoretical framework to describe it. Our work also fits into an existing line of work that empirically studies properties and behavior of neural networks, e.g. [2,3,4,5]. We believe that a theoretical understanding of why this phenomenon arises would be interesting, but given that we do not make theoretical claims, we do not believe that theoretical understanding should be a precondition for the publication of an empirical study.
>
> **References**
>
> [1] Nguyen, T., et al. "Do wide and deep networks learn the same things? uncovering how neural network representations vary with width and depth." International Conference on Learning Representations (2021).
>
> [2] Raghu, A., et al. (2019, September). Rapid Learning or Feature Reuse? Towards Understanding the Effectiveness of MAML. In International Conference on Learning Representations.
>
> [3] ​​Frankle, Jonathan, and Michael Carbin. "The lottery ticket hypothesis: Finding sparse, trainable neural networks." arXiv preprint arXiv:1803.03635 (2018).
>
> [4] Hermann, Katherine, and Andrew Lampinen. "What shapes feature representations? exploring datasets, architectures, and training." Advances in Neural Information Processing Systems 33 (2020): 9995-10006.
>
> [5] Gotmare, Akhilesh, et al. "A closer look at deep learning heuristics: Learning rate restarts, warmup and distillation." ICLR (2019).

---

### Author Response · Authors · 2022-09-22
**Summary of revisions**

We thank the reviewers for their time and feedback.

Given the reviewers’ comments, we have made the following changes to the manuscript (also highlighted in a different text color):

- Providing more clarification on the motivation of the investigation in Section 1

- Clarifying in our list of contributions (Section 1) as well as in Section 6 that our experimental findings suggest that regularizing the block structure has a minor impact on generalization

- Moving Figure 4, Figure 10, Figure 12, and Table 1 from the Appendix

- Adding Table 2 to report performance of Shake-shake regularization and transfer learning, compared to standard training

---

### Decision · Action_Editors · 2022-10-28

**Recommendation:** Accept with minor revision

**Comment:**

The paper has been reviewed by four reviewers. Three of them recommended "leaning accept" or "accept", while one recommended "reject".

The main concerns include the following. (1) Motivation and consequence. It is unclear what is the motivation of studying block-structure effect, and whether understanding this phenomenon can lead to improved training of deep networks. (2) Theoretical analysis. The paper is mostly empirical, without theoretical analysis. The authors pushed back on these two issues in their replies.

I think the phenomenon studied in this paper is interesting and noteworthy, and the origin of the phenomenon as revealed by the paper is convincing. So I lean towards accepting the paper.

In the revision, I hope the authors will continue to strengthen the part on motivation and consequence, and if possible, consider adding some theoretical understanding.

**Audience:**

Researchers who study the training of deep neural networks, either empirically or theoretically, may be interested in the findings of this paper.

**Claims And Evidence:**

The paper claims that block-structure effect is caused by dominant data points. The paper provides empirical evidence for this claim by training ResNet architectures on datasets such as CIFAR.

The phenomenon studied in this paper is curious and interesting. The experimental results are meaningful.